



**A review of methods for measuring groundwater-surface**
**water exchange in braided rivers**
Katie Coluccio[1], Leanne Kaye Morgan[1, 2]
[1]Waterways Centre for Freshwater Management, University of Canterbury, Private Bag 4800, Christchurch 8140,
New Zealand
[2]College of Science and Engineering, Flinders University, GPO Box 2100, Adelaide SA 5001, Australia
*Correspondence to:* Katie.coluccio@pg.canterbury.ac.nz





**Abstract.** Braided rivers, while uncommon internationally, are significant in terms of their unique ecosystems
and as vital freshwater resources at locations where they occur. With an increasing awareness of the connected
nature of surface water and groundwater, there have been many studies examining groundwater-surface water
exchange in various types of waterbodies, but significantly less research has been conducted in braided rivers.
Thus, there is currently limited understanding of how characteristics unique to braided rivers, such as channel
shifting; expanding and narrowing margins; and a high degree of heterogeneity affect groundwater-surface
water flow paths. This article provides an overview of characteristics specific to braided rivers, including a map
showing the regions where braided rivers are concentrated at the global scale: Alaska, Canada, the Japanese and
European Alps, the Himalayas and New Zealand. To the authors' knowledge, this is the first map of its kind.
This is followed by a review of prior studies that have investigated groundwater-surface water interactions in
braided rivers and their associated aquifers. The various methods used to characterise these processes are
discussed with emphasis on their effectiveness in achieving the studies' objectives and their applicability in
braided rivers. The aim is to provide guidance on methodologies most suitable for future work in braided rivers.
In many cases, previous studies found a multi-method approach useful to produce more robust results and
compare data collected at various scales. Ultimately, the most appropriate method(s) for a given study will be
based on several factors, including the scale of interactions that need to be observed; site-specific
characteristics; budget; and time available. Given these considerations, we conclude that it is best to begin
braided river studies with broad-scale methods such as airborne thermal imaging, differential flow gauging or
tracer analysis and then focus the investigation using finer scale techniques such as groundwater well
observations or temperature sensors. Given the challenges of working directly in braided rivers, there is
considerable scope for the increased use of remote sensing techniques and geophysics. There is also opportunity
for new approaches to modelling braided rivers using integrated techniques that incorporate the often-complex
river bed terrain and geomorphology of braided rivers explicitly. We also identify a critical need to improve
understanding of the role of hyporheic exchange in braided rivers; rates of recharge to/from braided rivers; and
historical patterns of dry and low-flow periods in these rivers.

**1    Introduction**
Until recently, groundwater and surface water systems were often considered separately both in research and in
the way they were managed as resources (Kalbus et al., 2006). However, understanding the interactions between
groundwater and surface water is now recognised as crucial to effective water resource management (Brodie et





al., 2007). These systems are connected, so the development or contamination of either groundwater or surface
water will often affect the other (Rosenberry and LaBaugh, 2008). Pumping from wells that are hydraulically
connected to surface water bodies can result in, for example, reduced flows in rivers or diminished lake levels,
or cause surface water inflow to groundwater (Stefania et al., 2018). Locations where groundwater and surface
water interact can serve as contaminant transport pathways (Chadwick et al., 2002). Groundwater seepage into
surface water can provide important nutrients and temperature regulation for aquatic organisms (Hayashi and
Rosenberry, 2002). Key questions in groundwater-surface water investigations are the location and flux of
groundwater discharge to surface water bodies, and conversely, surface water recharge to groundwater These
questions can be considered at various spatial and temporal scales (Lovett, 2015).

This article investigates the methods that have previously been used for examining groundwater-surface water
exchange in braided rivers. Braided rivers are a highly dynamic type of river with meandering channels, wide
bars and variable flow levels. Globally, braided rivers are relatively rare; they are mainly found in the Canadian
Rockies, Alaska, the Himalayas, New Zealand, and the European and Japanese Alps (Figure 1) (Tockner et al.,
2006). There are instances of braided rivers at locations outside of these regions (e.g., Russia, U.S., Scotland),
however these locations are not shown in Figure 1 because, at a global scale, they are not where braided rivers
are mainly found. Braided rivers generally occur in mountainous areas with a large sediment source (such as
glacial outwash), high river discharge rates and a steep topographic gradient (Charlton, 2008). These high-
energy environments enable the rivers to carry large sediment loads. When the rivers reach their capacity to
carry sediment, they form gravel braids, which branch out and re-join, creating gravel islands and shallow bars
(Figures 2 & 3). Bars and islands are often referred to as distinct features, with bars existing at periods of low
flow, while islands are generally more permanent features that may be vegetated (Charlton, 2008). Braided
rivers can completely change their geometry over a few decades. They undergo expansion and contraction
phases in which their channels widen or narrow, depending on sediment supply and river flows (Piégay et al.,
2006). The wetted channels of the river can shift, abandoning channels and re-occupying old channels
(Charlton, 2008). Relatively erodible streambanks, which allow for wide channels to form and meander, are a
key characteristic of braided rivers. These rivers generally have gravel beds but sand-bed rivers such as the
Brahmaputra-Jamuna, which begins in the Himalayas and flows through India and Bangladesh (and is the
world's largest braided river), can also form braided patterns (Sarker et al., 2014). The Brahmaputra-Jamuna is
the only braided river in this review that is not a gravel-bed braided river.





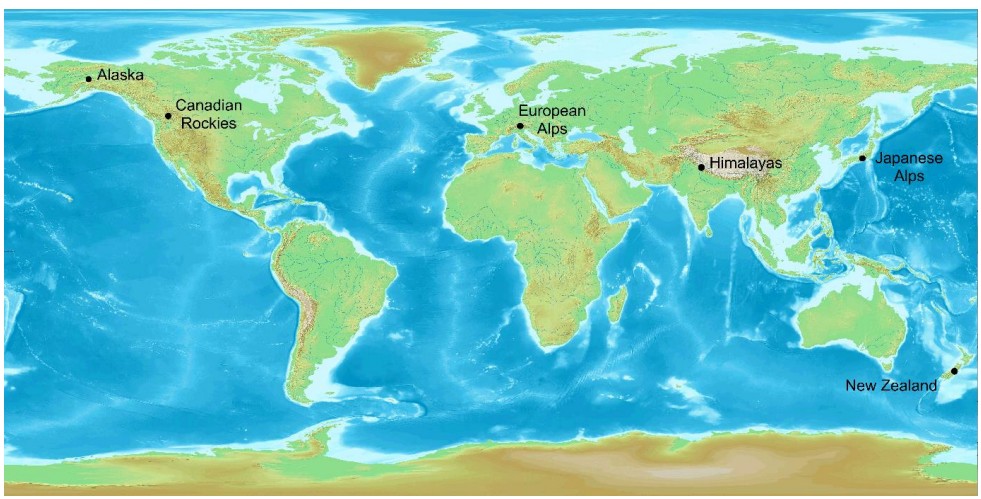


Figure 1. Locations where most braided rivers occur globally. Map base layer image attribution: "World Map-A
non-Frame" is licensed under CC BY-SA 3.0.

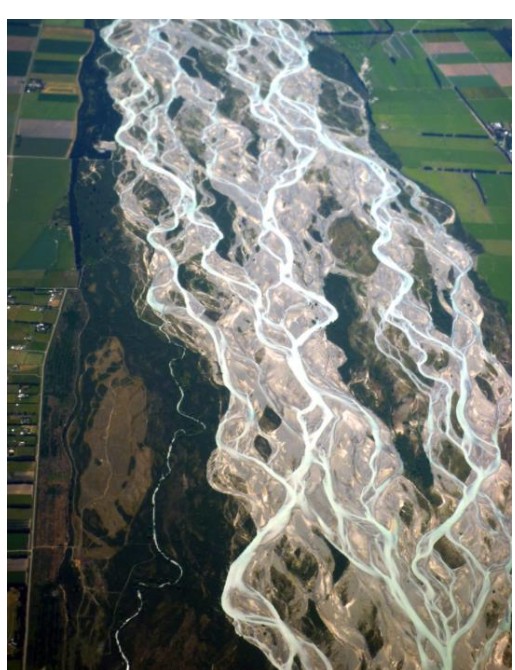

Figure 2. Rakaia River in New Zealand displaying a classic braided pattern. Image reproduced with permission
by Andrew Cooper.





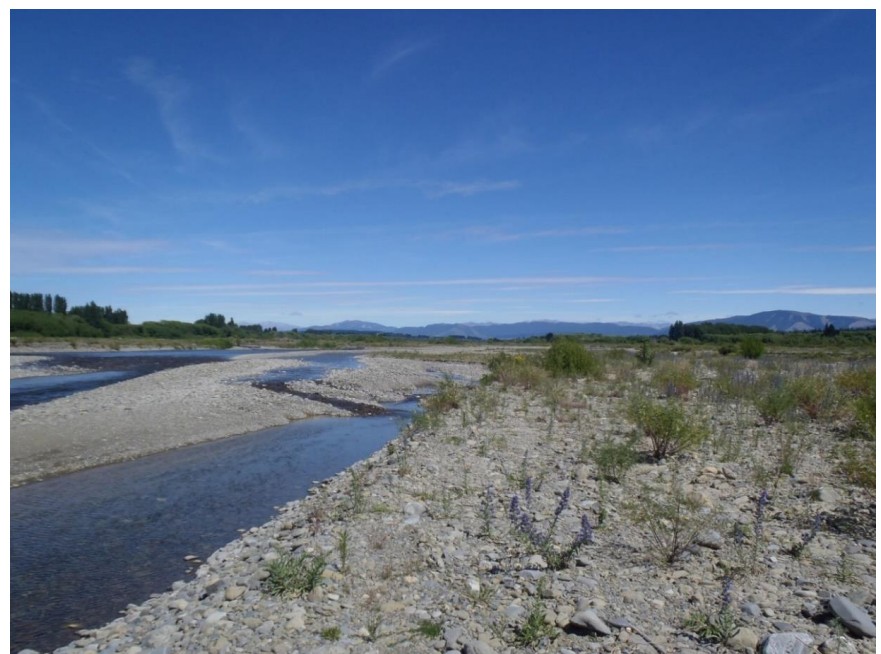


Figure 3. The Rakahuri/Ashley River in New Zealand displaying a typical braided river consisting of multiple
channels, gravel bars and vegetated islands. Photo: Katie Coluccio.

Braided river deposits have formed extensive aquifers throughout the world including many in the regions
shown in Figure 1 (Brown, 2001; Huggenberger and Regli, 2006). The complex depositional processes of
braided rivers create heterogeneous aquifer properties (Huggenberger and Regli, 2006), and a significant portion
of flow occurs in preferential flow paths formed by previous river flow channels (Close et al., 2014; Dann et al.,
2008; White, 2009). The complexity of braided rivers and their underlying heterogeneous aquifers makes
managing these systems in an integrated manner, that accounts for surface water-groundwater interaction,
challenging. For example, there is significant uncertainty surrounding rates of groundwater recharge from large
braided rivers in New Zealand, which complicates the sustainable allocation of water extraction rights from
surface water and groundwater sources (Close et al., 2014). There is also limited knowledge of how hyporheic
exchange in braided rivers affects groundwater-surface water interactions. In addition, braided rivers often have
reaches that become dry or have very low flows, and the historical patterns of these drying and low-flow
periods, and the impact of groundwater-surface water exchange on this, is an area of research where improved
knowledge is needed.






Braided rivers around the world have ecological, cultural, social, economic and recreational importance. Braided
rivers provide habitat for many plant and animal species specifically adapted to survive in the dynamic, nutrient-
poor environment of the rivers' gravel bars and their margins (Kilroy et al., 2004; Tockner et al., 2006). In New
Zealand, the rivers are some of the last remaining native habitat on the heavily modified Canterbury Plains of
the South Island, thus serving a vital ecological purpose for plant and animal species, many of which are
critically endangered (Caruso, 2006; Williams and Wiser, 2004). Braided rivers and their associated aquifers are
also important freshwater resources used for drinking water supplies, irrigation, stock water and hydropower. In
many areas, these rivers hold significant cultural, social and recreational value for their importance for food
gathering, boating and swimming, and as places of outstanding natural character.

However, braided rivers face pressure from many angles. In many places they are subject to damage from
vehicles, gravel extraction, invasive plant species, development on river margins, damming, low flow levels and
poor water quality (Department of Conservation [DOC], 2006). These factors can influence river processes in
many ways, including altering the rate of sedimentation or changing the flow regime, which may impact various
uses of these rivers, as well as riparian ecosystems (Piégay et al., 2006).

Much braided river research has focused on understanding their geomorphological structures and processes,
such as sediment transport (e.g., Ashmore, 1993; Chalov and Alexeevsky, 2015; Huggenberger and Regli, 2006;
Nicholas et al., 2006). Most studies up to the early 1990s consisted of laboratory-based modelling of the
braiding process (e.g., Ashmore, 1982; Young and Davies, 1991) and field studies of small reaches of valley-
confined systems (Ferguson et al., 1992). Beginning in the mid-1990s, there were advances in numerical models
to estimate the braiding process in reaches, remote sensing, and the quantification of river morphology and
morphological change using digital elevation models (e.g., Bernini et al., 2006; Copley and Moore, 1993;
Doeschl et al., 2006; Huggenberger, 1993). This allowed, for the first time, the visualisation and analysis of the
morphology of large braided rivers (e.g., Hicks et al., 2006; Huggenberger, 1993; Lane, 2006). A number of
studies have looked at the surface water features of braided rivers (e.g., Davies et al., 1996; Meunier et al., 2006;
Young and Warburton, 1996), as well as aquifers created by braided river deposits (e.g., Huber and
Huggenberger, 2016; Pirot et al., 2015; Vienken et al., 2017). However, the connections between the two have
been less explored. To highlight the scarcity of studies examining groundwater-surface water interaction in



braided rivers compared to other types of surface water bodies, a Web of Science search (on 8 February 2018)
for "groundwater and surface water interactions" in lakes, estuaries and small streams produced 437, 73 and 204
results, respectively, compared to only six results for braided rivers (note that this database search does not
reflect *all* studies conducted in braided rivers, only what was found in this key word search, but this
significantly smaller number of results highlights the relative scarcity).

This article addresses this gap in the literature by reviewing methods previously used in braided rivers
internationally to characterise groundwater-surface water interactions. The objective is to provide guidance for
future braided river studies. As described in this section, braided rivers have many unique features, which may
make it difficult to apply techniques used in different river environments. In relation to the methods used in
previous studies, this article examines the equipment and study design; cost; issues of temporal and spatial
scales; and ultimately the techniques' effectiveness. For general overviews of methodologies not specific to
braided river applications, refer to Kalbus et al. (2006); Brodie et al. (2007); Rosenberry and LaBaugh (2008);
and Lovett (2015).

**2    Methodologies for assessing groundwater-surface water interactions in braided rivers**
Various types of methods have been used to investigate groundwater-surface water exchange in braided rivers
such as mass balance approaches; natural and artificial tracers; direct measurement of hydraulic properties; and
modelling. Many of these studies employed multiple methods to meet their objectives. To thoroughly and
clearly assess each method, the techniques, and their advantages and limitations will be discussed individually in
the following section, and the discussion section will review the merits and limitations of multi-method studies.
This information is then summarised in Table 1.

**2.1    Water budgets**
Some of the most commonly used methods for identifying gains and losses to braided rivers have been based on
a mass balance approach. The underlying principle of this method is that any gain or loss of surface water can be
related to the water source, therefore the groundwater component can be identified and quantified (Kalbus et al.,
2006). Many of these mass balance approaches have used water budgeting to separate groundwater and surface
water components both on river-reach and catchment-wide scales.





### 2.1.1 River-reach water budgets

River-reach water budgets involve estimating the net flux of seepage in a defined river reach by measuring

stream discharge in cross sections and then calculating the difference in flow between the cross sections (Kalbus

et al., 2006). If there is an increase or decrease in discharge, this can be considered as a gaining or losing reach,

respectively, provided any surface inflows or outflows (e.g., tributary inflows, abstractions) are accurately

quantified. Measurements should generally be taken in low-flow conditions to eliminate the influence of recent

rainfall (Brodie et al., 2007).

Several studies on the South Island of New Zealand have used river-reach water budgets to identify gaining and

losing reaches of braided rivers. The Selwyn River, which has losing and gaining reaches, and annually dries in

parts, has been the focus of several studies (Larned et al., 2008; Larned et al., 2015; Vincent, 2005). Larned et

al. (2008) investigated the patterns of losing and gaining along the river by flow gauging at 19 sites between two

permanent flow recorders on the river. The authors used this gauging data to classify gaining and losing reaches

of the river as perennial, ephemeral or intermittent depending on the percentage of time that the river flowed and

the source of the flow (i.e., ephemeral reaches in this study were sourced from runoff only, whereas intermittent

reaches were groundwater sourced when the water table intersected the river channel and could also have a

runoff component) (Larned et al., 2008). Larned et al. (2008) also compared flow at two sites on the river with

data from 11 nearby groundwater wells for a five-year period to assess lag times between the two systems. In a

subsequent study on the Selwyn River, Larned et al. (2015) used a 30-year gauging record from two flow

recorder sites on the river to calculate groundwater level lag times. Vincent (2005) conducted an in-depth study

of the hydrogeology of the upper Selwyn River catchment, which included concurrent flow gauging to estimate

gains and losses from the river to groundwater. In another study, Farrow (2016) characterised gaining and losing

reaches of the four major rivers in the Ashley-Waimakariri zone using historic flow gauge records. The author

cited the need for additional concurrent gauging under mean flow conditions to more accurately characterise

long-term gaining and losing reaches. In an attempt to determine the causes of the perennial drying of the North

Branch of the Ashburton River, Riegler (2012) conducted flow gauging along the river in conjunction with

groundwater well measurements, mapping of dry reaches and regression analysis. Despite the various methods

used, the cause of drying in the North Branch could not be determined (Riegler, 2012). Burbery and Ritson

(2010) incorporated various methods such as flow gauging, piezometer surveys, hydrochemical sampling and

stable isotope analysis with catchment water use data to characterise the groundwater and surface water





interactions in the Orari River catchment on the South Island of New Zealand. The authors used the flow
gauging data to classify gaining and losing reaches in four of the rivers in the catchment. They agreed with
previous conclusions by Davey (2004) that in order to obtain a greater level of detail about groundwater-surface
water connectivity at the local scale, shorter spaced flow gauging coupled with high-resolution piezometric
surveys and aquifer pumping tests should be carried out (Burbery and Ritson, 2010). In a 2012 study of the
Waimakariri River in the Canterbury region of New Zealand, White et al. (2012) used a steady-state
groundwater budget to estimate groundwater outflow from the riverbed based on the mean daily flow at a
recorder site on the river between 1967-2009 and seven-year groundwater level observations in a monitoring
well array beside the river. The authors found that river channel area rather than channel position was most
important in their calculations. However, they recommended that future research examine the effects of channel
position and area on groundwater outflow from the Waimakariri River. This is particularly relevant in braided
rivers, as their channel positions often change.

Flow gauging has also been used outside of New Zealand to investigate groundwater-surface water interactions
in braided rivers. Both Simonds and Sinclair (2002) and Doering et al. (2013) used flow gauging as part of
multi-method studies for estimating groundwater-surface water interactions in the Dungeness River
(Washington State, U.S.) and Tagliamento River (northeastern Italy), respectively. These authors conducted
concurrent gauging to calculate the net loss or gain of flow along river reaches and compare to data collected
from other methods.

**2.1.2  Catchment-scale water budgets**
Some studies have used catchment-scale water budget calculations to estimate the inflow and outflow from
braided river catchments and distinguish groundwater from surface water sources. The underlying relationship is
provided below (modified from Scanlon et al. (2002)):
$inflow = outflow \pm \Delta S$        (1)
Here, inflow is the sum of precipitation, surface water inflow and groundwater inflow. Outflow is comprised of
actual evapotranspiration, surface water outflow and groundwater outflow. $\Delta S$ is the change in water storage in
the catchment. This also considers artificial changes to water levels in the catchment such as industrial
discharges to surface water or water abstraction.



Burbery and Ritson (2010) calculated a water budget for the Orari River catchment in Canterbury, New Zealand
to aid in characterising groundwater-surface water interactions in the catchment. This model was based on field
observations from various methods including flow gauging and groundwater well observations, climate data and
water use data. The authors concluded that the model provides a good basic understanding of the Orari
catchment, but a sensitivity analysis of the model should be carried out. They also recommended additional
investigation of the deep groundwater system to better understand its hydraulic connection to shallow
groundwater, as the authors believed deep upwelling groundwater might be supplying shallow groundwater.

In another study in Canterbury, New Zealand, Anderson (1994) calculated a regional groundwater budget for the
area between the braided Selwyn and Rakaia rivers, which included a consideration of inflows and outflows to
these rivers. Inputs to the water budget were rainfall, recharge from surface water estimated by flow gauging,
sea water intrusion into the aquifer, inflow from other aquifers, leakage from stock water races and artificial
recharge (i.e., land surface recharge from irrigation). Outputs were groundwater abstraction, groundwater-fed
spring flow, river baseflow, groundwater discharge to the sea, flows to other aquifers and evapotranspiration.
This water budget provided a useful indication of flows in and out of the groundwater system in the study area,
but it is important to note that there were significant uncertainties with some parameters. For example, there
were large uncertainties with river loss rates, and this study was done before groundwater abstraction was
metered in this region (so actual use was not known).

**2.1.3  Advantages and Limitations**
Reach-scale water budgets are useful for identifying hotspots of river gains and losses at a broad scale.
Streamflow gauging provides more reliable information in relatively homogeneous environments (which
braided rivers often are not, as discussed below).

There are several issues regarding the effectiveness of river-reach water budgets for characterising groundwater-
surface water interactions in braided rivers. As detailed in Section 1, these types of rivers are typically
comprised of heterogeneous materials and thus there may be small-scale interactions of groundwater and surface
water within reaches, of which flow gauging is poor at identifying (Hughes, 2006). For example, Larned et al.
(2015) noted that lag time calculations can only highlight generalised flow paths, whereas predicting more
specific groundwater flow paths or residence times would require studies using additional techniques such as





tracers or potentiometric data. Also, accurate measurements of flow rates can be compromised by several factors
including interference of macrophytes in the streambed, low flow, imprecise or shifting river margins, high
sediment load, or unstable streambeds that permit parafluvial flow (i.e., flow in the area of riverbed that is to
some extent annually scoured by flooding (Stanford, 2007)). As noted by Close et al. (2014), there is significant
uncertainty around estimates of river to groundwater flows solely based on hydraulic measurements, particularly
for large braided rivers, as these environments provide various challenges for accurate flow measurements.
These systems are difficult to measure because precise flow gauging can only be carried out during low flows
and measurement errors can be considerable (Close et al., 2014). In larger rivers, often the measurement error is
greater than the net exchange of groundwater and surface water (LaBaugh and Rosenberry, 2008).

Catchment water budgets can be a useful method at a larger scale but are generally not appropriate for assessing
small-scale groundwater-surface water interactions, as the accuracy of recharge rates to or from rivers is limited
by the accuracy of the measurement of the other components in the budget (Scanlon et al., 2002). They can be
simple and quick to calculate, but this depends on how time consuming or expensive the data collection is. Also,
this method can have low resolution because of the limited number of flow gauging stations on rivers (Kalbus et
al., 2006). Thus, when calculating budgets for large catchments, the errors can be significant.

**2.2    Environmental tracers**
There are various natural physical and chemical properties of groundwater and surface water that can serve as
indications of interaction between the two systems. A variety of tracers have been used in braided rivers to
investigate groundwater-surface water exchange including geochemical tracers such as conductivity, chloride
and alkalinity; stable isotopes; and radioactive isotopes such as radon. At sites where there is a discernible
difference between the groundwater and surface water concentrations of one of these parameters, the influence
of groundwater or surface water may be able to be detected. This type of analysis assumes there is evenly
distributed groundwater discharge or recharge between sampling locations and that there is complete mixing of
water sources (Lovett, 2015). To separate surface water or groundwater components, mixing models based on
conservation of mass are used (Kalbus et al., 2006), such as End Member Mixing Analysis (EMMA) or
hydrograph separation. The methods presented below represent the majority of known braided river applications
to date, and thus this is not a complete list of all tracers used in previous studies. Some additional tracers applied
in braided river settings not discussed here include dissolved oxygen (e.g., Larned et al., 2015; Rodgers et al.,



2004), silica (e.g., Botting, 2010; Rodgers et al., 2004; Soulsby et al., 2004), nitrate (e.g., Burbery and Ritson,
2010; Larned et al., 2015; White et al., 2012) and sulphate (e.g., Acuña and Tockner, 2009; Botting, 2010).

**2.2.1   Stable isotopes**
Oxygen, which is a key component of water, naturally occurs in two stable isotopic forms: oxygen-16 ($^{16}O$) and
oxygen-18 ($^{18}O$) (Sharp, 2007). Due to the difference in mass between the two isotopes, they undergo
fractionation during evaporation and condensation (Taylor et al., 1989). The process is largely driven by
temperature and elevation, whereby precipitation is increasingly depleted in $^{18}O$ at higher elevations and colder
temperatures (Sharp, 2007). The ratio of $^{16}O$ to $^{18}O$ (referred to as $\delta^{18}O$) is used to identify the relative
concentrations of the two isotopes. This allows for the identification of groundwater recharged by alpine sources
and lowland rainfall (Burbery and Ritson, 2010) and can shed light on groundwater flow paths in aquifers.

Several studies have used $\delta^{18}O$ to characterise groundwater-surface water exchange in New Zealand's braided
rivers and associated aquifers. Blackstock (2011) analysed $\delta^{18}O$ and $\delta D$ ($\delta D$ refers to the ratio of deuterium, a
stable isotope of hydrogen) to identify groundwater flow paths and recharge sources in the Christchurch
groundwater system. Blackstock (2011) found that the isotopic model matched well with previous physical mass
balance calculations for the groundwater system and that stable isotope analysis was useful, especially in
shallow groundwater, while recommending future studies of the system combine both stable isotopic and
physical hydrological methods (e.g., piezometric surveys). Botting (2010) carried out another stable isotope
study, examining groundwater flow patterns and origins on the north bank of the braided Wairau River in the
Marlborough region. In his multi-method study including $\delta^{18}O$ and $\delta D$ analysis, hydrochemical sampling, pump
tests, and groundwater well observations, he found that stable isotope analysis was the most effective technique
for distinguishing surface water from groundwater. Vincent (2005) aimed to establish the relationship between
surface water and groundwater in the upper Selwyn River catchment using $\delta^{18}O$ analysis, flow gauging and
groundwater well observations. The $\delta^{18}O$ analysis enabled the identification of groundwater recharge sources.

Burbery and Ritson (2010) used $\delta^{18}O$ analysis to determine alpine versus lowland recharge sources for
groundwater in the Orari River catchment in New Zealand. Of the various methods used in the study (which also
included flow gauging, a catchment-scale water budget, chemical tracers and groundwater level observations),
the authors found $\delta^{18}O$ analysis to be highly effective for understanding groundwater-surface water interactions



in the catchment. Given $\delta^{18}O$ varies seasonally, they recommended sampling be carried out at various times during the year to obtain better temporal resolution. Additionally, long-term sampling would allow climatic variations to be considered. The authors also noted that to accurately understand water mixing in a catchment, $\delta^{18}O$ sampling must consider all end members (i.e., surface water, rain water, soil water and groundwater).

In a regional study of the depth and spatial variation of groundwater chemistry on the central Canterbury Plains in New Zealand, Hanson and Abraham (2009) carried out $\delta^{18}O$ and other hydrochemical analyses along two transects across the plains. The authors found $\delta^{18}O$ to be the most reliable tracer to differentiate between land surface recharge and alpine river water on the Canterbury Plains. However, they pointed out that a suite of tracers would be needed to characterise groundwater flow paths and groundwater recharge sources. They also noted that $\delta^{18}O$ can be significantly altered where alpine water is used for irrigation.

**2.2.2  Radon**

Radon is another useful tracer for identifying groundwater-surface water interactions. Radon-222 (Rn-222), a chemically and biologically inert radioactive gas, is part of the Uranium-238 decay process and is present in nearly all rocks and soils (LaBaugh and Rosenberry, 2008). As water flows through rocks and soils it becomes enriched in Rn-222. In surface waters, radon quickly degasses, so groundwater generally has Rn-222 concentrations three to four orders of magnitude higher than surface waters, thus making it an effective tracer in many environments (Burnett et al., 2001). For example, an area of high radon concentrations in surface water would suggest groundwater inflow. It is a cost-effective, simple technique that is suitable for studying large areas (Martindale, 2015).

Rn-222 analysis can address many questions related to groundwater and surface water interactions. In a multi-method study in the braided Tagliamento River in northeast Italy, Acuña and Tockner (2009) used Rn-222 to assess the residence time of upwelling groundwater in the hyporheic zone. Moore (1997) analysed radon and barium concentrations in the Bay of Bengal in Bangladesh at the mouth of the Brahmaputra River. In the Bay of Bengal, sediment deposited by the Brahmaputra River provides a significant source of radon and barium. This sediment is mainly deposited during high flows. Moore (1997) found that radon and barium concentrations were also high during low flows and concluded that this is due to groundwater inflow to the river.





Close et al. (2014) assessed the effectiveness of Rn-222 analysis to characterise surface water recharge from the
Waimakariri River to groundwater in the Canterbury Plains in New Zealand. While it is feasible in most cases to
obtain in-stream water samples to measure groundwater inflow to surface water, sampling surface water outflow
to groundwater is more complicated. A well network of sufficient size is needed to enable sampling of shallow
groundwater at a suitable distance from the river (within 2 to 3 weeks of groundwater transport time) (Close et
al., 2014). Close et al. (2014) used Rn-222 sampling to calculate the velocity of groundwater recharge in the
aquifer using the ingrowth (i.e., the rate of build-up in a closed system) equation for Rn-222. Radon
concentrations were measured in shallow groundwater wells near a reach of the river known to lose flow and
compared with radon concentrations in the river. The study did not include a calculation of recharge fluxes, and
the authors noted that to do this would require several known parameters (or assumptions), including
dimensions of the recharge area, whether recharge is constant along a particular reach and the effective porosity
of the groundwater system. The authors also noted that estimations of groundwater velocities from wells located
at regular intervals down a river could shed light on spatial variations in recharge volumes, which may help
avoid uncertainties around estimating aquifer dimensions and properties. The authors also recommended that a
high-resolution study with closely spaced sampling sites could be useful for highlighting preferential flow paths
in the riparian zone (e.g., channels of open framework gravel).

In another New Zealand-based study, Close (2014) sampled Rn-222 in the Wairau River in Marlborough and in
groundwater wells within five kilometres of the river to better understand the groundwater-surface water
interactions in the river and the amount and variability of recharge to the groundwater system. The author also
measured temperature, dissolved oxygen and pH, as these physicochemical properties are often distinct in
groundwater and surface water. Close found that temperature correlated well with the spatial distribution of the
radon. The author noted some recommendations for future radon studies including that the samples should be
counted for longer during analysis (referring to liquid scintillation counting (LSC)), to reduce the analytical
percentage error, which increases at low radon concentrations. Close (2014) also added that there could be
significant errors with estimating groundwater flow paths due to local heterogeneity and the meandering nature
of the alluvial deposition process in the area. Close (2014) recommended analysing temperature and data
collected from piezometers in conjunction with radon to resolve these uncertainties.





There are some limitations of Rn-222 analysis, as it requires several assumptions, including that stream water is
well mixed downstream of groundwater discharge areas; water fluxes are constant; the radon activity in the
stream water and groundwater are known and constant; and that there is no additional surface recharge from
sources such as streams or stock water races (Kraemer and Genereux, 1998). It also may be difficult to
distinguish between regional groundwater discharge and hyporheic zone exchange using radon analysis (Lovett,
2015; Martindale, 2015).

**2.2.3   Chloride**
The chloride ion (Cl⁻) can be used as an indicator for groundwater and surface water mixing in locations with
sufficiently distinct chloride concentrations in groundwater and surface waters. Surrounding the braided Bow
River, which flows from the Canadian Rocky Mountains through the province of Alberta, the groundwater has
elevated levels of chloride from road salting. Cantafio and Ryan (2014) measured chloride levels in an urban
reach of the Bow River to assess water quality impacts and baseflow sources for the river. They found that
nearly all river flow originates in the Rocky Mountains and there is little contribution from groundwater.

Chloride is frequently sampled amongst a suite of hydrochemical parameters to investigate groundwater and
surface water interactions, as groundwater often becomes enriched in chloride as it passes through soil and rocks
(Dommisse, 2006). Burbery and Ritson (2010) measured chloride concentrations in the Orari River catchment in
New Zealand, specifically looking at chloride-to-sulphate ratios to delineate groundwater-surface water
interactions and examine recharge sources in the catchment. They found that basic ion chemistry was useful for
determining the extent of the Orari River water but noted that results can be complicated by hydrochemical
changes due to land use activities. Acuña and Tockner (2009) also analysed chloride in the Tagliamento River in
northeast Italy to better understand groundwater and surface water interactions. The authors used EMMA to
determine the relative proportion of freshly infiltrated surface water in upwelling hyporheic zone water. Larned
et al. (2015) also measured chloride along with several hydrochemical parameters to correlate groundwater and
surface water mixing in the Selwyn River in New Zealand. Botting (2010) analysed chloride ions in his study of
groundwater flow patterns and origins on the North Bank of the Wairau River in the Marlborough region of
New Zealand. Similarly, Domisse (2006) analysed chloride in order to characterise recharge sources on the
Hinds-Rangitata plain in Canterbury, New Zealand.





### 2.2.4 pH and alkalinity


pH and alkalinity can serve as effective indicators for determining catchment water sources. In a study of the
River Feshie, a braided river in the Cairngorms in Scotland, Rodgers et al. (2004) used alkalinity as a tracer to
investigate temporal changes in stream water hydrochemistry and characterise sources of river flow. They
collected in-stream samples fortnightly for one year and at a finer resolution during two rainfall events in this
period. The authors noted that Gran alkalinity is particularly useful as it serves as a directly measurable, close
approximation to the acid neutralising capacity, which is considered a conservative chemical tracer. Gran plots
are commonly used to determine alkalinity and acid neutralising capacity in water with low alkalinity or low
conductivity. A Gran function plot identifies the point at which all alkalinity has been titrated in a strong acid-
strong base titration (USGS, 2012). Rodgers et al. (2004) used EMMA to estimate different hydrological
sources of River Feshie water. The authors were reasonably confident of their estimates because of the extensive
temporal and spatial components of their study. Because of the relative simplicity and low cost of the Gran
alkalinity method, these types of longer term and detailed spatial surveys are becoming increasingly feasible
(Rodgers et al., 2004), though may be costly in terms of human resources required. They also measured pH and
correlated it with alkalinity levels in the stream water.

In another study in the Feshie catchment, Soulsby et al. (2004) conducted a geochemical tracer study to improve
large-scale flow path understanding in this 231 km$^2$ catchment. The authors carried out chemical-based
hydrograph separations to separate baseflow from storm event sources. They conducted flow gauging on a
fortnightly basis for two years and used 10 years of flow data in their analysis. They collected water samples on
a wide spatial scale at median flow levels. They analysed for pH and Gran alkalinity amongst other parameters.
They found these were simple and inexpensive to measure. These methods have proven useful in the United
Kingdom (UK) as they distinguish between water sourced from acidic, organic soils (which are common in the
UK at shallow depths) and deep, older groundwater. Soulsby et al. (2004) found their study provided valuable
information at the sub-catchment scale, but more information was needed at finer spatial scales.

### 2.2.5 Advantages and Limitations


Environmental tracers can provide significant insight into both catchment-wide hydrology, as well as provide
estimations of seepage flux on the point scale (Close, 2014; Dommisse, 2006; Lovett, 2015). Even considering
catchment heterogeneity, some tracers can behave predictably enough to serve as effective tracers for studies of



braided rivers (Soulsby et al., 2004). Environmental tracers are useful in settings where there is a sufficient
difference between tracer concentrations in the groundwater and surface water, and some tracers can be easily
incorporated in long-term routine monitoring programs. Disadvantages of these methods include that
hydrochemistry of the baseflow and storm event water composition may be too similar, or that hydrochemistry
may not be constant in time or space (Genereux and Hooper, 1998). Also, land use activities may alter
hydrochemistry in catchments, for example from fertiliser application or irrigation (Soulsby et al., 2004).
Additionally, some low tracer concentrations may cause analysis errors (e.g., in the case of radon) (Close,

441    2014).


### 2.3    Heat tracers

Temperature has been used in a number of studies, particularly in Europe, to characterise groundwater-surface
water interactions in braided rivers. In most locations, during winter and summer months, there is a discernible
difference in groundwater and surface water temperatures. In general, groundwater temperature is more stable,
whereas surface water temperatures change diurnally and seasonally (Kalbus et al., 2006). In summer,
groundwater is typically colder than surface water, whereas in winter, groundwater is generally warmer. Heat
tracer methods can be used to identify discharge and recharge zones as well as quantify the flux of water moving
between groundwater and surface water systems (Andersen, 2005). There are various methods involving
temperature sensing that range in complexity, scale and cost. One-off temperature readings can be taken using
probes, or sensors or data loggers can gather time-series data both in-stream or in groundwater wells. Vertical
and horizontal temperature profiles can also be measured by arranging sensors in a series either in-stream or in
wells on river margins. The studies discussed below demonstrate various applications of temperature
measurement to characterise groundwater-surface water exchange.

Tonolla et al. (2010) investigated thermal heterogeneity in the braided river floodplains of the Roseg River in
Switzerland and the Tagliamento River in Italy. They used thermal infrared imaging to identify surface
temperature patterns at 12 to 15-minute intervals over 24-hour cycles. They took photos using an infrared
camera with a measurement accuracy of ±0.5°C set up on a tripod at the rim of the mountains bordering the
catchment. This allowed for higher spatial and temporal resolution images to be captured than if taken aerially.
However, the authors noted that the large zenith angles of the images may have impacted the temperature
measurements and thus would need to be taken into consideration. The authors found that thermal infrared





imagery is a powerful, non-invasive tool for understanding thermal heterogeneity in complex river systems.
They also measured the vertical temperature distribution at 3 to 5-minute intervals in the top layer of unsaturated
gravel deposits (1 cm spacing, 0-29 cm depth) on the two floodplains using thermocouples attached to PVC
frames, which were buried in the sediment. They found that the temperature differences in the top 29 cm of
unsaturated sediment varied almost as much as thermal variation across the entire floodplain.

In another study of the Tagliamento River, Acuña and Tockner (2009) investigated how groundwater and
surface water interactions affect temperature changes in the river, a key factor in the health of ecosystems.
Along four reaches of the river, they continuously measured temperature during the summer of 2007 and the
winter of 2007-2008. They used other methods such as mini-piezometers to determine hydraulic gradient, and
radon and chloride analysis to shed light on groundwater-surface water mixing. They found that there was
minimal groundwater discharge to the river reaches, but they did find that there was considerable hyporheic
zone upwelling, which influenced surface water temperatures and provided potential refuge for aquatic
organisms.

Malard et al. (2001) investigated thermal heterogeneity in the Roseg River in Switzerland. They monitored
surface water temperatures and measured the temperature in sediments using mini-piezometers (at 30 and 80 cm
deep) for one year. They found that the direction and magnitude of surface water-groundwater exchanges
significantly influenced the vertical pattern of water temperature (Malard et al., 2001). The authors also found
that the groundwater source (e.g., shallow alluvial, deep or hillslope) resulted in very different effects on
seasonal temperature changes in the hyporheic zone.

In another European Alps study, Passadore et al. (2015) conducted thermal monitoring to characterise the
temporal and spatial variability of streambed water fluxes in the Brenta River in Italy. They used heat as a tracer
in conjunction with water level measurements. Passadore et al. (2015) found this combination of methods to be
effective in estimating groundwater-surface water interactions. They measured temperature and water levels
both in the stream and on the riverbank using piezometers. They noted that this method requires continuous
monitoring of water temperature and levels of the stream and groundwater.





Two studies of the Wairau River in Marlborough, New Zealand analysed temperature (Close, 2014; Close et al.,
2016). Close (2014) measured temperature in the river and in groundwater wells located near the river to
characterise river recharge to the aquifer. The author compared the data to Rn-222 analysis and found that the
measured temperatures correlated well with the spatial distribution of radon. Close et al. (2016) logged
temperature in 17 groundwater wells in 15-minute intervals, using the daily mean temperature values to estimate
the lag time between the river and the observation wells. Close et al. (2016) used the average monthly
temperatures in the wells as an input for a numeric model. They found that only qualitative conclusions could be
drawn due to the relative nature of the recharge estimates (Close et al., 2016).

Lastly, Coluccio (2018) demonstrated, for the first time, the use of diurnal temperature signal analysis to
characterise seepage through the streambed of a braided river. Temperature probes with a series of evenly
spaced temperature sensors were installed into the riverbed of the Ashburton River on the South Island of New
Zealand. The study determined the direction and magnitude of vertical seepage through the streambed. The
results were compared with chemical analysis and water level measurements in the river and shallow
groundwater to better inform the interpretation of the temperature data. Coluccio (2018) found that it was
difficult to distinguish between shallow groundwater and hyporheic flow. The author also noted that further
studies would benefit from combining a point-scale method like temperature probe analysis with broader scale
techniques (Coluccio, 2018).

**2.3.1  Advantages and Limitations**
Heat tracers offer many techniques at varying spatial and temporal scales. Broad-scale methods like aerial
thermal imaging can be used to obtain large-scale data, and they can offer the advantage of remote collection of
data in areas that are difficult to access. Point-scale techniques using temperature sensors on the other hand can
indicate surface water-groundwater interactions at a specific location. Some methods of temperature analysis
can also quantify seepage flux (e.g., diurnal signal analysis, (see Coluccio, 2018)). The methods range in cost
and complexity, and thus can be tailored to suit a study's needs. There are some limitations including that a
temperature gradient between groundwater and surface water might not always be present (e.g., this may be
affected by environmental conditions such as wind, shade from vegetation or rapidly changing river levels)
(Johnson, 2003). Also, for certain types of analysis, temperature needs to be measured continuously (Irvine et
al., 2017).



### 2.4 Hydraulic property measurement

Several studies have directly measured hydraulic properties to quantify flow between groundwater and surface water in braided rivers (e.g., Acuña and Tockner, 2009; Botting, 2010; Dommisse, 2006; Malard et al., 2001; Shu and Chen, 2002; Simonds and Sinclair, 2002). This has included direct groundwater level measurements in wells (Aitchison-Earl and Ritson, 2013; Doering et al., 2013; Dommisse, 2006; Simonds and Sinclair, 2002; Vincent, 2005) and various well tests such as slug and pumping tests (Botting, 2010; Chen, 2007; Coluccio, 2018).

#### 2.4.1 Groundwater well observations

Groundwater levels are often used to aid in the understanding of groundwater-surface water interactions, and there have been several studies conducted in braided rivers using this technique. Groundwater level data can be used to identify the hydraulic gradient (i.e., the difference in hydraulic head over a given distance) at a location, which can reveal groundwater discharge to a river and river recharge into an aquifer. The underlying principle is that if groundwater levels in a well are higher than the river level, the river is gaining (i.e., because groundwater is flowing into the river). Conversely, where river levels are higher than the groundwater level in a nearby well, the river is losing (i.e., because river water is flowing into groundwater).

Hydraulic gradient is calculated as $\Delta h/\Delta l$, where $\Delta h$ [L] is the difference in hydraulic head [L] and $\Delta l$ is the distance between the points where the hydraulic head was measured. Hydraulic gradient can be measured in the horizontal direction to characterise flows into or out of a river through the sides of the river. Here, $\Delta h$ [L] is the difference between the groundwater level in a well at the edge of the river and a well a distance $\Delta l$ [L] away from the edge of the river. Hydraulic gradient can also be measured in the vertical direction, to characterise flows into or out of the river through the river bed. In this case, $\Delta h$ [L] is the difference between the groundwater level in an in-river piezometer and the river level at that location; and $\Delta l$ [L] is the distance from the riverbed to the top of the well screen (Doering et al., 2013).

Once the hydraulic gradient has been measured, the magnitude of groundwater flow into or out of a river can be estimated using the Darcy equation:

$$Q = -KA\frac{\Delta h}{\Delta l} \tag{2}$$





Where $Q$ [L$^3$/T] is the volume of flow; $A$ [L$^2$] is the cross-sectional area perpendicular to flow through which the
water passes; and $K$ [L/T] is hydraulic conductivity (Schwartz and Zhang, 2003). For calculating the horizontal
flow magnitude, a horizontal hydraulic conductivity of the surrounding aquifer is generally used. To calculate
the vertical magnitude of flow, the vertical hydraulic conductivity of the streambed needs to be determined as
does the streambed area over which the water exchange occurs (Simonds and Sinclair, 2002).

In terms of specific methods that can be used for measurements, existing groundwater wells or piezometers near
rivers can be useful for conducting these types of studies, particularly given the high cost of drilling new wells.
Mini-piezometers, which are scaled-down versions of piezometers and typically installed no deeper than about
two metres (Figures 4 & 5), have been previously used in studies of braided rivers (Acuña and Tockner, 2009;
Doering et al., 2013; Malard et al., 2001).

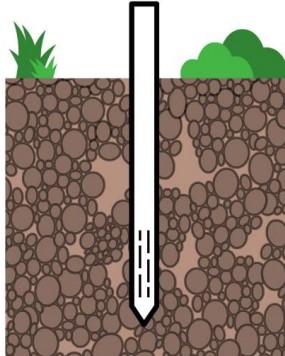


Figure 4. Conceptual diagram of a mini-piezometer. Image source: Steve Coluccio



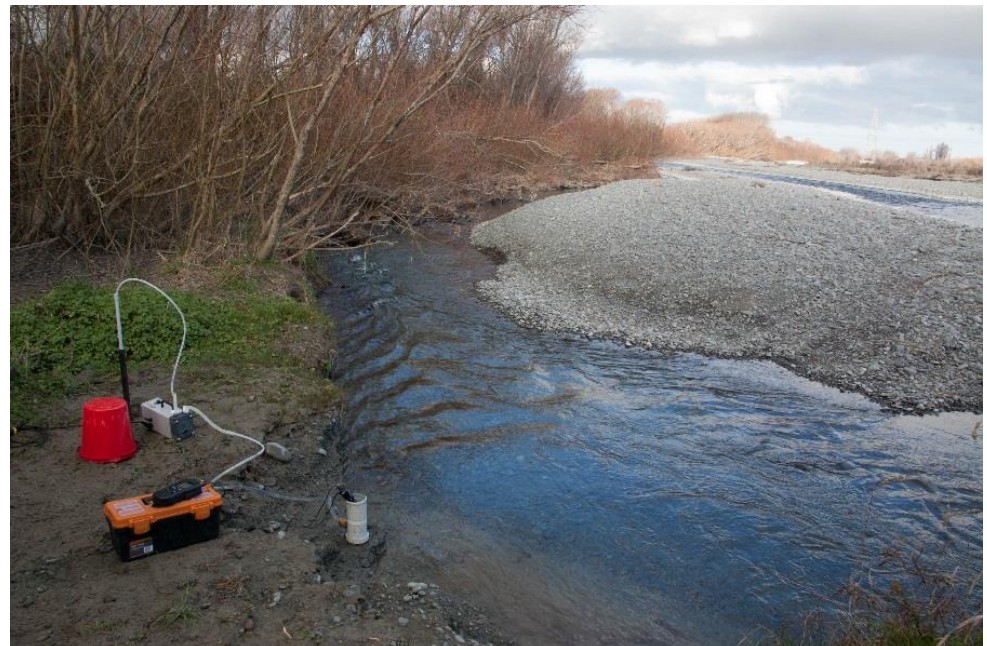


Figure 5. Mini-piezometer installed on the bank of a braided river. Image source: Katie Coluccio

Previous studies have examined the correlations between groundwater levels and river levels to establish the
degree of connectedness of groundwater systems and braided rivers, for example, attempting to identify the
causes of drying reaches and changes in long-term river flows. Prior studies have been carried out in catchments
with substantial agricultural surface and/or groundwater abstraction for irrigation. Thus, the questions here are
often whether abstraction has caused drying in rivers or decreases in river flows, and what effect future
abstraction will have. These studies have often coupled groundwater level measurements with streamflow
gauging and physicochemical sampling of river water and groundwater. Riegler (2012) explored these questions
in the North Branch of the Ashburton River in Canterbury, New Zealand, which has annually become dry in
recent years. Riegler (2012) examined groundwater levels, in conjunction with flow gauging, to attempt to
correlate groundwater levels and decreased flow levels in the river. The study concluded that there were too
many uncertainties, particularly around the complex behaviour of the groundwater system, to draw strong
conclusions on the causes of the drying riverbed. Several other studies also investigated New Zealand braided
rivers that are highly connected to groundwater using these methods (Aitchison-Earl and Ritson, 2013; Larned
et al., 2008; Larned et al., 2015; Vincent, 2005; Coluccio, 2018).





A multi-method study was carried out on the Dungeness River in Washington State in the United States to
characterise groundwater-surface water interactions. Simonds and Sinclair (2002) installed 27 mini-piezometers
in the river in which they measured the vertical hydraulic gradient between the stream and water table. They
also continuously monitored water levels and temperature in two well transects, providing data on the horizontal
hydraulic gradient and temporal changes in groundwater-surface water flows. The authors also conducted flow
gauging along "seepage runs" in the river to quantify the net gain or loss of flow over a reach. This information
was used to calibrate a model used to predict the impacts of land use change in the catchment.

Groundwater level measurements in mini-piezometers have also been applied in studies of European braided
rivers. Malard et al. (2001) calculated the difference in hydraulic head between hyporheic water and surface
water in their study of the Roseg River in Switzerland. They used a manometer in the piezometers to measure
the water levels and calculate the vertical hydraulic gradient in piezometers installed in the river. For
piezometers installed in the parafluvial zone, they calculated horizontal hydraulic gradient. Acuña and Tockner
(2009) also incorporated groundwater level observations into their multi-method study of the Tagliamento River
in Italy. The used PVC mini-piezometers installed to a depth of 50 cm in four reaches of the river. They
calculated vertical hydraulic gradient to determine the direction and intensity of surface and subsurface (i.e.,
hyporheic flow or groundwater) exchange in the streambed. In another study of the Tagliamento River, Doering
et al. (2013) installed mini-piezometers along 10 transects in losing and gaining reaches of the river. Five mini-
piezometers were installed horizontally across the river at each location and were used to calculate the vertical
hydraulic gradient where the piezometers were installed.

**2.4.2   Hydraulic conductivity tests**
As detailed above, the hydraulic conductivity of riverbeds is needed to calculate the magnitude of flow through
the riverbed. There have been a number of studies investigating the hydraulic conductivity of streambeds (e.g.,
Landon et al., 2001; Kelly and Murdoch, 2003), though few studies have been conducted in braided rivers.
There are many well-established methods for calculating hydraulic conductivity of a porous medium, including
grain size analysis, permeameter tests, slug and bail tests, and pumping tests (see Fetter, 2001).

In an early investigation of the permeability of gravel streambeds, Van't Woudt and Nicolle (1978) extracted
gravel from the bed of the braided Waimakariri River in Canterbury, New Zealand. They conducted lab-based



tests to determine hydraulic properties of the bed substrate such as porosity and infiltration rates. This study
resulted in several conclusions about sub-surface flow in gravel-bed rivers including that fine sediments flowing
through the gravels tend to create a low-permeability clogging layer along the margin of and below the riverbed.
Interestingly, the authors also found horizontal permeability to be far higher than vertical permeability (30:1),
but it is difficult, if not impossible, to draw conclusions about horizontal and vertical conductivities once the
sediment is disturbed.

Cheng et al. (2010) carried out a study to determine the statistical distribution of streambed vertical hydraulic
conductivity at 18 sites along a 300-km reach of the Platte River in Nebraska. They conducted in-situ
permeameter tests using falling head tests and found that vertical hydraulic conductivity was normally
distributed at all but one of their study sites.

In a study on the North Bank of the Wairau River in Marlborough, New Zealand, Botting (2010) conducted
pumping tests (along with stable isotope and hydrochemical analysis, geomorphological mapping, and
groundwater level observations) to determine groundwater flow paths and origins. The pumping tests were of
limited use however, because the pumping did not successfully lower the groundwater levels, most likely due to
the high transmissivity of the aquifer.

On the Ashburton River in New Zealand, Coluccio (2018) conducted slug tests in mini-piezometers installed on
the margins of the river. Due to the high permeability of the sediments, a pneumatic slug testing device
(Michell, 2017) was used so that the water level in the wells could be effectively lowered and the removal of the
slug could be carefully controlled. Air was pumped into the well to lower the water level and then
instantaneously removed to begin the rising head test. A data logger that took more than one reading per second
was necessary to record a sufficient number of data points for analysis. The hydraulic conductivity values
calculated from the slug tests were on the low end of the range for expected hydraulic conductivity values in this
area, which may have been a reflection of the tests being conducted in localised areas of finer sediments,
highlighting the limits of using this point-scale method in heterogeneous environments (Coluccio, 2018).

**2.4.3   Advantages and Limitations**





There are various benefits and drawbacks of the methods described in this section. Use of existing groundwater
wells may be very useful in a study, but the installation of new wells generally comes at a high cost. Mini-
piezometers offer an inexpensive and simple method for obtaining groundwater level and pressure data (Lee and
Cherry, 1978). They are easy and quick to install in most locations, and the analysis of their measurements is
generally straightforward (Brodie et al., 2007). They can be used in small-scale applications and in detailed
surveys in heterogeneous environments (Fritz et al., 2016). However, measurements at a study site must be
taken at the same time to be representative of similar flow conditions (Kalbus et al., 2006). In previous studies,
groundwater level observations have rarely been used in isolation and typically have been coupled with other
methods.

The heterogeneous composition of braided rivers complicates the estimation of the hydraulic conductivity of
streambeds on a reach or catchment scale. Hydraulic conductivity can vary significantly across an area, even
with small changes in sediment composition, thus it is difficult to extrapolate values to represent a large area
(Brodie et al., 2007). With grain size analysis, the structure and stratification of the sediment are destroyed
during analysis, so the conductivity value does not represent the vertical or horizontal conductivity (Cheng and
Chen, 2007). Similarly, when conducting permeameter tests it is difficult to transport sediment samples without
disturbing their structure (Kalbus et al., 2006). However, these tests can be used as a primary estimation before
conducting further tests. Slug tests are quick and simple to carry out and a significant advantage is that they only
require one well. Pumping tests on the other hand require a pumping well and an observation well, which can be
cost prohibitive. Pumping test results provide average hydraulic conductivity values across a larger area than for
slug tests, thus their results may be less sensitive to heterogeneous conditions (Kalbus et al., 2006), whereas slug
tests provide information only about the location where the well is installed.

**2.5   Modelling**
Computer modelling is often used for the estimation of exchange between surface water and groundwater as a
complement to field measurements. Such computer models have become irreplaceable tools to gain insight into
real-world surface water-groundwater issues ranging from system understanding at the local or regional scale to
future projections for management purposes. The complexity of numerical hydrological models used for this
purpose range from simple conceptual models that treat subsurface compartments (i.e., groundwater) as
reservoirs where inflows or outflows are specified, to highly complex integrated models that have a more





realistic physical coupling between surface water and groundwater. MODFLOW (Harbaugh, 2005) is the most
commonly used numerical model to simulate surface water-groundwater interaction (Furman, 2008; Barlow and
Harbaugh, 2006). As pointed out by Wöhling et al. (2018), MODFLOW is considered to be a good compromise
between integrated and conceptual modelling approaches. Several packages are available within MODFLOW
for simulating surface water-groundwater interaction including the River Package (RIV) (McDonald and
Harbaugh, 1984), the Stream Package (STR1) (Prudic, 1989), the Streamflow-Routing Package (SFR1) (Prudic
et al., 2004), and the Streamflow-Routing 2 Package (SFR2) (Niswonger and Prudic, 2005). Common to all of
these packages, is that flows to or from the river are calculated as the product of streambed hydraulic
conductance (a lumped parameter summarising the geometry of the river and the clogging layer, which in
practice generally acts as a calibration parameter), and the difference between the hydraulic head in the river and
hydraulic head of the groundwater. In the case where the groundwater head is below the base of the river (i.e.,
the groundwater and river are disconnected), flows from the river are the product of the hydraulic conductance
and the difference between the hydraulic head of the river and the elevation of the river bed bottom. Further
details of the application and limitations of the MODFLOW packages can be found in Brunner et al. (2009) and
Brunner et al. (2010) and will not be repeated here.

While the modelling of braided rivers is not new, it has been done more often from a geomorphological
perspective (e.g., Ashmore, 1993; Copley and Moore, 1993; Meunier et al., 2006). Nevertheless, a number of
published studies detail modelling of braided rivers for the purposes of understanding flow dynamics and
pumping impacts. Shu and Chen (2002) developed a transient MODFLOW model to understand the spatial and
seasonal variation in surface water-groundwater exchange for the Platte River, in Nebraska, one of the most
well-known braided rivers in the United States. Shu and Chen (2002) also simulated pumping from over 1000
wells to better understand the relationship between seasonal pumping from irrigation wells and river levels.
Simulation results suggested that continued over-extraction of groundwater in the region would gradually
increase losses from the river. In a subsequent study, Chen (2007) developed a numeric model using the
Galerkin finite-element method to assess the impact of the riparian vegetation on baseflow interception and
increased river water infiltration in the Platte River.

Passadore et al. (2015) developed a three-dimensional groundwater flow model for the large and complex
Central Veneto aquifer in northeast Italy, which is highly connected to the braided Brenta River. The authors



calibrated the model using historic observation data from groundwater wells. They first created a conceptual
geologic model, which was used to set up a finite-element numeric model. The used aquifer parameter inputs
(e.g., hydraulic conductivity, transmissivity) based on field observations and carried out a steady-state
calibration based on observations from 100 wells. The authors provide no detail regarding the manner in which
the Brenta River was implemented in the model but note that fluxes from the river to the underlying aquifer are
an important component of aquifer recharge.

There are several braided river studies conducted in New Zealand that involved modelling groundwater-surface
water exchange. In the Canterbury region, Anderson (1994) and Scott and Thorley (2009) developed relatively
simple steady-state MODFLOW models for the purpose of improving understanding of regional-scale water
budgets, which include gains and losses from the braided Rakaia, Selwyn and Waimakariri Rivers. In Hawkes
Bay, Baalousha (2012) used MODFLOW to characterise groundwater-surface water interactions in the
Ruataniwha Basin, which contains several braided rivers. The modelling indicated that the braided rivers in this
region gain much more than they lose. Wilson and Wöhling (2015) attempted to improve the understanding of
Wairau River recharge into the Wairau aquifer in Marlborough. They used flow gauging and groundwater level
observations to calibrate a steady-state MODFLOW model. The river was modelled using the SFR2 package.
The authors noted groundwater monitoring records and pump testing showed the aquifer to be more complex
and stratified than previously thought, indicating that groundwater monitoring sites were likely only
representative of local conditions. This finding further highlights the difficulties of modelling highly
heterogeneous, complex river systems and their associated aquifers. Close et al. (2016) used the Wilson and
Wöhling (2015) MODFLOW model as a basis for a study using heat as a tracer in the Wairau aquifer. Close et
al. (2016) simulated thermal transport using MT3DMS in MODFLOW, which is incompatible with the SFR
package, used in the original model, so they used the STR package. They compared results in the flow model to
their thermal transport model and found that the transport model produced higher recharge rates on average and
that it did not provide "unique insight" into the model parameters, unlike the flow model, which was based on
river flow and groundwater levels. Close et al. (2016) also found that model calibration fit the observed flow
and groundwater levels well but not the observed groundwater temperatures, indicating the aquifer was more
heterogeneous than captured in the model. This heterogeneity was an important consideration to fit the model to
the temperature data. In a subsequent study of the Wairau Plain aquifer and the Wairau River, Wöhling et al.
(2018) developed a transient MODFLOW model that was calibrated using targeted field observations as well as





"soft" information from experts of the local water authority. The uncertainty of simulated river-aquifer exchange
flows was evaluated using Null Space Monte Carlo methods. The study suggested that the river is hydraulically
perched (losing) above the regional water table in its upper reaches and is gaining in the downstream section. It
was found that despite large river discharge rates (i.e., regularly reaching 1000 m$^3$/s), the net exchange of flow
rarely exceeded 12 m$^3$/s and seemed to be limited by the physical constraints of unit-gradient flux under
disconnected rivers. An important finding for the management of the aquifer was that changes in aquifer storage
are mainly affected by the frequency and duration of low-flow periods in the river.

**2.5.1   Advantages and Limitations**
Field methods are often time consuming and expensive, and they may not be at the targeted spatial or temporal
scale. Therefore, the estimation of exchange between braided rivers and groundwater is often complemented by
hydrological modelling. MODFLOW is commonly used to model surface water-groundwater interaction,
including in braided rivers. Complex flow channel geometry, which changes over time, is not explicitly
incorporated into modelling efforts, at least in the studies identified by the authors listed above. As such, the
impact of complex and temporally variable flow channel geometry on surface water-groundwater exchange is
not well understood. More complex integrated modelling approaches than that possible using the MODFLOW
suite of packages is likely required to incorporate this level of detail. A future integrated approach that considers
channel geometry in a more physically realistic manner may be facilitated by the recent development of braided
river terrain models (e.g., Williams et al., 2016).

**3    Discussion**
There are many factors to consider when selecting the appropriate method(s) for groundwater-surface water
investigations, and there are special considerations relevant to braided river environments. The most appropriate
method will depend on physical and hydrological conditions in the setting and scale of interaction to be
measured (LaBaugh and Rosenberry, 2008). As a result of this review of studies investigating groundwater-
surface water exchange in braided rivers, a summary table has been developed (Table 1) that outlines the
literature discussed in this paper and the advantages and disadvantages of the various methods used in these
studies.



**Table 1.** Advantages and disadvantages of various methodologies for estimating groundwater-surface water interactions in braided rivers

| Method | Advantages | Disadvantages | Applications of these methods in braided rivers* |
|---|---|---|---|
| River-reach water budgets | • Suitable for relatively homogeneous aquifers<br>• Useful for identifying hotspots of river gains and losses | • Errors can be greater than the amount of groundwater-surface water flux<br>• Not well suited for sub-reach scale<br>• Not very accurate in highly heterogeneous systems<br>• Does not consider streambed throughflow<br>• Multiple sites on a river must be gauged concurrently | Acuña & Tockner (2009); Aitchison-Earl & Ritson (2013); Burbery & Ritson (2010); Doering et al. (2013); Farrow (2016); Larned et al. (2008); Larned et al. (2015); Riegler (2012); Simonds & Sinclair (2002); Soulsby et al. (2004); White et al. (2012); Williams & Aitchison-Earl (2006) |
| Catchment-scale water budgets | • Good for large-scale studies<br>• Simple and can be relatively quick | • Errors can be significant in large catchments<br>• Uncertainties of land surface recharge and offshore flow rates can result in errors<br>• Can be expensive and time consuming depending on how data is collected | Anderson (1994); Burbery & Ritson (2010) |
| Environmental tracers (e.g., radon, stable isotopes, chloride, etc.) | • Good for environments where there is a sufficient difference between tracer concentrations in groundwater and surface water<br>• Useful for identifying interactions at small and large scales<br>• Some tracers can easily be included in long-term, routine sampling (e.g., pH, alkalinity, dissolved oxygen)<br>• Can be used to quantify seepage rates | • Analysis errors can be an issue when concentrations are low (e.g., radon)<br>• Groundwater and surface water concentrations may be too close to differentiate<br>• Land use activities may cause hydrochemical changes<br>• Concentrations may not be temporally or spatially consistent | Acuña & Tockner (2009); Blackstock (2011); Botting (2010); Burbery & Ritson (2010); Cantafio & Ryan (2014); Close (2014); Close et al. (2014); Coluccio (2018); Doering et al. (2013) Domisse (2006); Guggenmos (2011); Larned et al. (2015); Malard et al. (2001); Moore (1997); Rodgers et al. (2004); Soulsby et al. (2004); Vincent (2005) |
| Heat tracers | • Variety of methods ranging in complexity, cost, scale<br>• Can be used for both locating areas of discharge/recharge and quantifying flux<br>• Aerial surveys can be faster than in-stream surveys | • Often needs to be measured continuously<br>• Need a sufficient temperature difference between groundwater and surface water<br>• May be less effective in periods of high river flows | Acuña & Tockner (2009); Close (2014); Close et al. (2016); Coluccio (2018); Doering et al. (2013); Lovett et al. (2015); Malard et al. |



| | | | |
|---|---|---|---|
| | | | (2001); Passadore et al. (2015); Tonolla et al. (2010) |
| Modelling | • Acts as a database for field data<br>• Can assist researchers to develop intuition about physical processes and refine their conceptual models<br>• Useful for carrying out regional-scale assessments for management purposes, such as determining streamflow depletion associated with pumping<br>• MODFLOW packages widely accepted for numerical simulation and intuitive to apply<br>• MODFLOW packages considered a good compromise between a simple conceptual modelling approach and a more complex integrated approach | • Some models have high computational and time requirements<br>• Various assumptions required that may not reflect actual hydraulic processes or aquifer properties | Anderson (1994); Baalousha (2012); Chen (2007); Close et al. (2016); Passadore et al. (2015); Scott & Thorley (2009); Shu & Chen (2002); Wilson & Wöhling (2015); Wöhling et al. (2018) |
| Hydraulic Property Measurement | • Mini-piezometers are easy and quick to install<br>• Wells can be installed in-stream or on land<br>• Can be used in small-scale or regional applications<br>• Can be used to survey heterogeneous areas<br>• Piezometer measurements are straightforward to analyse | • Deep groundwater wells are expensive to install<br>• All measurements at a study site must be taken at the same time<br>• Hydraulic conductivity can significantly vary spatially, thus difficult to extrapolate to represent a large area | Acuña & Tockner (2009); Aitchison-Earl & Ritson (2013); Botting (2010); Burbery & Ritson (2010); Chen (2007); Cheng et al. (2010); Coluccio (2018); Doering et al. (2013); Domisse (2006); Larned et al. (2008); Larned et al. (2015); Malard et al. (2001); Riegler (2012); Shu & Chen (2002); Simonds & Sinclair (2002); Van't Woudt & Nicolle (1978); Vincent (2005); Williams & Aitchison-Earl (2006); Wilson & Wöhling (2015); Wöhling et al. (2018) |

*Note: some studies referenced in this table were not discussed in the text.





The objectives of the study will influence which methods are most applicable. If only qualitative information
about groundwater-surface water exchange is required, this could be obtained by methods such as mapping the
locations of wet and dry reaches of a river, or identifying where there is mixing between groundwater and
surface water based on chemical or heat tracers. Alternatively, if quantitative data is needed, such as the rate of
groundwater seepage into a surface water body, this may be obtained by measuring Rn-222, analysing
temperature signals, or by calculating the hydraulic gradient. Researchers have developed flux quantification
techniques for some of the methods discussed in this paper (e.g., for temperature analysis see Gordon et al.,
2012), but it is important to consider inputs required to calculate seepage through a streambed, such as
streambed hydraulic conductivity (see section 2.4). If direct water samples are needed, tools to consider could
include groundwater wells or piezometers. Water samples and flux rates can also be obtained using seepage
meters, a common method used for estimating groundwater-surface water interactions typically based on the
design proposed by Lee (1977). However, it does not appear that these devices have been previously used in
gravel-bed braided rivers. Seepage meters have various limitations as discussed in previous studies (e.g., Kelly
and Murdoch, 2003; Brodie et al., 2009; Cey et al., 1998), which indicate their application in braided rivers
would be difficult and less effective than other methods.

It is important to match the scale of the data required with the methods being used. This should include the
consideration of both spatial and temporal scales. If regional or catchment-scale information is desired, methods
such as pumping tests, flow gauging, stable isotope analysis, solute tracers and chemical analysis are among the
most applicable methods. Remote sensing techniques such as airborne thermal infrared imaging and geophysics
may also prove useful to apply in braided river settings for gathering data on a large scale, as these methods
have been used in braided rivers for geomorphological studies (e.g., Huber and Huggenberger, 2016) and for
investigating groundwater-surface water exchange in other settings (McLachlan et al., 2017). It is important to
recognise that it may be difficult to accurately characterise groundwater-surface water interactions in highly
heterogeneous environments based on broad-scale methods. At the reach scale, oxygen-18 or radon analysis
could be appropriate methods (Lovett, 2015). At a point scale, streambed piezometers and temperature profiles
can be useful. With finer resolution methods, there may be issues with up-scaling the data because many closely
spaced measurements are needed, and it is difficult to distinguish between groundwater discharge and hyporheic
zone flow (Lovett, 2015). While point-scale data may be desired, it may be impractical to carry out the large
number of measurements necessary on a wider scale (such as in a large river). Using a combination of broad and



point-scale techniques at a single study site may help overcome the limitations of the individual techniques,
particularly in heterogeneous environments (Kalbus et al., 2006). Temporal scale variabilities are also important
to consider. The magnitude and direction of groundwater-surface water interactions may change in response to
factors such as river flow levels (Rosenberry and LaBaugh, 2008). Some methods may require that all sampling
be completed within a short time period so that the data is representative of similar conditions. For instance,
concurrent flow gauging, where the flow in reaches on a river are gauged on the same day, will generally
produce a more reliable representation of baseflow conditions compared to gauging carried out over multiple
days, as flow levels can change daily (Farrow, 2016). Temperature profiling on the other hand may need to be
continuous over a period of time to remove the influence of diurnal fluctuations (Passadore et al., 2015)
depending on the method of analysis.

Conceptualisation and quantification of hydrogeological systems is generally associated with a degree of
uncertainty. The degree of accuracy of measurements can vary based on many factors including sampling
protocol, lab analysis, assumptions required (e.g., aquifer properties) or the nature of the method chosen. The
degree of accuracy dictated by the study objectives should be carefully considered when choosing the
appropriate methods. Likewise, the level of accuracy and confidence in results should be discussed in
conjunction with study results.

Site-specific characteristics will largely determine the most appropriate methods to use. The geology,
topography, hydrochemistry, hydrology and hydrogeology of the study site will need to be considered. Factors
such as geologic complexity, chemical components of the soils and surface and ground waters, aquifer
properties, and climate should be taken into account. Inputs and outputs to groundwater and surface water may
need to be considered, such as abstraction for irrigation or industrial discharges. There are various practical
considerations such as the availability of groundwater wells, river access and feasibility of techniques. For
example, large braided rivers with high flows and deep channels may prove difficult to access directly. There is
also a reasonable risk of the loss or damage of equipment installed in braided riverbeds due to floodwaters or
sediment movement during storms. These practical considerations underline the potential benefits of remote
techniques to collect data in this type of river.



As with any study, the available resources will influence the types of methods selected. Techniques vary in cost
depending on materials needed, installation requirements or analysis methods. Mini-piezometers, for example,
are on the inexpensive end of this range, while airborne thermal imaging is a more expensive method, though its
cost may be reduced by using Unmanned Aerial Vehicles. Time is a key consideration, and this can range
widely. While simple and relatively inexpensive, some field techniques, such as streamflow gauging or
piezometer measurements, may be time consuming to carry out given the large number of measurements
required to obtain a representative sample, especially in heterogeneous environments like braided rivers. If
many replicate samples are required to obtain representative data for an area, it may be cheaper to use remote
sensing or another broad-scale method. Analysis requirements should be considered when evaluating the merits
of particular methods. Some chemical sampling for example may require expensive lab analysis and then
subsequent statistical analysis, whereas other methods such as flow gauging require minimal processing of data.
The availability of data relevant to the study site will be important to consider. For example, aquifer properties
may need to be known to carry out calculations or modelling. Or, historical sampling records may be needed to
compare long-term trends.

Despite these various considerations involved in choosing the appropriate methods for carrying out
investigations of groundwater-surface water interactions, according to Landon (2001), the number of
measurements made may be more important for obtaining accurate data than the type of methods chosen given
the spatial variability in hydraulic conductivity of streambeds. Also, as demonstrated in the various studies
discussed in this review, rarely did researchers rely on a single method to explore groundwater-surface water
interactions. As Kalbus et al. (2006) conclude in their comprehensive review of methodologies, the most
accurate results for estimating fluxes between groundwater and surface water may be achieved by combining
multiple methods at various scales. A multi-method approach may also help overcome the challenges of
working in heterogeneous braided river environments. Indeed, most of the studies presented in this article used
more than one technique to investigate questions relating to groundwater-surface water exchange.

**3.1    Key gaps and possibilities**
This paper has highlighted that there are currently gaps in the knowledge of how groundwater and surface water
interact in braided rivers. One of the most significant gaps in our understanding relates to hyporheic exchange.
We have limited understanding of how hyporheic flow processes operate, and how they impact river flow levels



and water quality in braided rivers. The hyporheic zone has been highlighted as a significant area for ecological
processes in rivers (Febria et al., 2011; Malard et al., 2001), but as Kalbus et al. (2006) note, it can be difficult to
differentiate between hyporheic exchange and groundwater discharge. In addition, despite the contributions of
the studies discussed here, the recharge rates to and from braided rivers continue to be a source of question for
water scientists and managers, as these have implications for both water quality and quantity. Measuring
seepage rates is still difficult in many gravel-bed braided rivers, and often there is significant uncertainty in the
data collected. Lastly, there is still much scope for research on identifying historical patterns of dry and low-
flow periods in braided river reaches. This is often an area of significant concern for communities that are
seeking answers on the correlations between dry or low-flow periods, and current and historical water use
practices and climate.

There is also room for improvement in the methods available to carry out these investigations. Refinement of
techniques that allow for direct measurements of physical or chemical properties in braided rivers would be
helpful. While the studies presented here have employed some direct methods, there is still a need for techniques
that can be used in braided rivers with coarse gravel substrate, fluctuating flow levels, and shifting channels and
gravel bars. Methods that can better capture the heterogeneous properties of braided rivers would be ideal.
Previous studies discussed in this paper have shown the promise of using environmental tracers such as Rn-222
and stable isotopes, as well as heat tracers in these settings. Given the challenges of working directly in braided
rivers, there is considerable scope for the use of remote techniques, such as thermal infrared imaging and
geophysics, to collect data on these rivers. As discussed in the modelling section of this paper, there is also
opportunity for new approaches to modelling of braided rivers.

**4  Summary**
Braided rivers are unique and dynamic river environments that serve important ecological, cultural, recreational
and freshwater resource functions. A critical aspect of their effective management is understanding groundwater
and surface water interactions in these rivers and their associated aquifers. This article provides an overview of
characteristics specific to braided rivers, which include multiple meandering channels that often shift; temporary
and semi-permanent bars and islands; heterogeneous and (typically) gravel streambeds; and dynamic flow
levels. We present a map showing the regions where braided rivers are concentrated at the global scale: Alaska,
Canada, the Japanese and European Alps, the Himalayas and New Zealand. To the authors' knowledge, this is





883 the first map of its kind. Our review of prior surface water-groundwater interaction studies in braided rivers

884 showed that most studies have been recent (in the past 10-20 years) and they have investigated a range of

885 questions including calculating seepage rates to/from braided rivers; estimating time lags between rivers and

886 groundwater; and looking at the implications of groundwater-surface water exchange on ecological processes.

887 We also investigated the effectiveness of the various methods used in the studies identified in this review in

888 terms of achieving the studies' objectives and their applicability in braided rivers. A table has been produced

889 summarising these findings and shows that there is a variety of available methods ranging in cost and scale.

890

891 Lastly, this article explored the various considerations one may make when choosing appropriate techniques for

892 investigating groundwater-surface water exchange in braided rivers. While the methods selected will ultimately

893 depend on a number of factors (e.g., budget and time requirements; spatial and temporal scales; data inputs

894 required; and site-specific characteristics), we conclude that the most effective approach will likely involve the

895 initial use of broad-scale approaches such as airborne thermal imaging, differential flow gauging, catchment

896 water budgets or tracers. Finer scale methods such as groundwater well observations, small-scale tracer studies

897 and temperature sensors can then be used to explore hot spots of exchange or specific areas of interest. The use

898 of multiple methods at varying spatial scales at a single study site may help overcome the uncertainties

899 associated with data gathered in heterogeneous braided river environments. Given the challenges of working

900 directly in braided rivers, there is considerable scope for the increased use of remote sensing techniques and

901 geophysics. There is also potential for new approaches to modelling braided rivers using integrated techniques

902 that incorporate the often-complex riverbed terrain and geomorphology of braided rivers explicitly. There is

903 presently limited understanding of the role of the hyporheic zone in surface water-groundwater exchange in

904 braided rivers, recharge rates to and from braided rivers, and historic drying and low-flow trends in braided

905 rivers; thus future research is needed in these areas.


907 **Author contribution**

908 The project was instigated by LM. KC carried out the literature review that formed the content of this

909 manuscript and wrote the initial manuscript draft. KC and LM revised the manuscript together.


911 **Acknowledgements**





This work was supported by the Canterbury Regional Council and the Waterways Centre for Freshwater
Management. Their support is greatly appreciated. The authors would like to thank Zeb Etheridge and Philippa
Aitchison-Earl for comments on an early version of this manuscript.



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
