# Peer review of "A review of methods for measuring groundwater-surface"

_Hydrology and Earth System Sciences, 2018_

## Referee Comment (RC1) · Anonymous Referee #1 · 31 Jan 2019

The manuscript "A review of methods for measuring groundwater-surface water exchange in braided rivers" by Katie Coluccio and Leanne Kaye Morgan is a review paper. As the title suggests it is about measuring methods for groundwater-surface water exchange in braided rivers. In general, the manuscript is informative, provides an overview about the current literature, is well structured and well written. However, some sections are lengthy and might be shortened. Furthermore, as indicated in the major comments below important information, definitions, etc. is missing. In general, the authors could think a little bit more out of the box. They are very focused on the methods that have already been used in studies of groundwater-surface water interactions in braided rivers. But there are several similar groundwater-surface water

interfaces and as part of a scientific review paper I would expect the authors to consider additional methods that might be adapted to braided rivers in future in addition to simply summarizing the literature available at present. I think the manuscript can be published after revision.

MAJOR COMMENTS:

===============

Entire manuscript: Try to shorten your manuscript and avoid lengthy descriptions of the literature, e.g. L173-L213, L216-242, L289-325, L328-L379, L382-402, L533-603, L606-L640, L667-L739.

L60 & entire manuscript: Suggest also methods that have been successfully used at other groundwater-surface water interfaces and that might be adapted to braided rivers and might be used in braided rivers in future. Reporting only what has already been done in braided rivers is a little bit thin.

L64 & Fig. 1 & L882: I strongly recommend adding all additional instances of braided rivers outside of the major regions. You might use different symbols for major regions with braided rivers and single instances.

L100f; L791f, L855: I think there is a need for clear definitions of "groundwater-surface water interactions" and of "hyporheic exchange". Often, the term "groundwater-surface water interaction" is used in literature in a wide sense including hyporheic exchange as one process of groundwater-surface water interactions. However, according to line 100f you consider both as separate processes with some impacts on each other.

L134ff: Even though I agree that there is little research about groundwater-surface water interactions in braided rivers your "Web of Science" search is meaningless. I tried to reproduce it. First of all "groundwater and surface water interactions" with "…" results in much smaller numbers than the ones reported by you, e.g. only three papers for lakes instead of 437 reported by you. Repeating the search without "…" resulted in approximately the numbers reported by you. However, having a closer look at those papers revealed that most of the hits are not about groundwater-surface water interactions at all but that the separate words of the phrase are used in separate sentences and in different context. Furthermore, at many of the interfaces mentioned by you (lakes, ocean, stream) specific terms are used, e.g. "lacustrine groundwater discharge", "submarine groundwater discharge" and "hyporheic zone" instead of "groundwater and surface water interactions". Sometimes the word "interactions" is substituted by "exchange" or by "interfaces". Also, there are different spellings for "groundwater" such as "ground water". I am quite sure that the largest number of studies focusing on groundwater-surface water interactions is about stream, followed by (coastal) oceans followed by lakes and finally by braided rivers. You might also have a look at review papers focusing on the different interfaces. There are several of them. I recommend either deleting lines 134-139 or repeating this literature search with a set of different keywords to get a more comprehensive overview of the literature of interest.

L158ff: From my experience budgets are often quite error-prone because accurate measurements of river discharge are challenging. Often changes in river discharge between stations are much smaller than the error inherent to the measurements. You should mention this shortcoming more clearly than only in lines 261-263.

L272ff/L284ff: I think it is important to introduce here also the concept that tracers need to be conservative (on the scale of the investigation). In this context, I doubt that dissolved oxygen (L284), nitrate (L285), sulphate (L286) and pH (L404) are useful tracers. pH might be acceptable in the context of alkanity but that also needs more discussion. The concentrations of oxygen, nitrate, sulfate and H+ will be altered due to many different biogeochemical processes. They might be used under certain circumstances and on small scales on which little turnover takes place. But this is something very critical. If you list these compounds you need to discuss them critically.

L272ff: In addition to environmental tracers I recommend to discuss also artificial tracers that might be added to the system. There are multiple studies using artificial tracers

and I am quite sure that the also have been used in braided rivers. However, even if not they are an option that should be considered.

L457-468: I don't see any connection of this paragraph to the topic groundwater-surface water interactions. Therefore, I recommend deleting this paragraph.

L469-484: The topic of the present review is measurement methods for groundwater-surface water interactions. Thus, these two paragraphs don't fit to the topic of the review paper. They are about impacts of groundwater and surface water on temperature (and ecological consequences) but not how to use measurements to identify groundwater-surface water interactions.

L502ff: I think it is important to measure temperature depth profiles as you do in this paragraph. However, you should go into a little bit more detail here and also mention typical evaluation methods for temperature depth profiles such as the steady state approach (e.g. C. Schmidt, M. Bayer-Raich, and M. Schirmer. Characterization of spatial heterogeneity of groundwater-stream water interactions using multiple depth streambed temperature measurements at the reach scale. Hydrology and Earth System Sciences 10:849-859, 2006) or VFLUX.

L443ff: I think at one point in this subchapter you should clearly differentiate between methods that are used to determine fluxes (e.g. temperature depth profiles) and methods for pattern identification (aerial TIR, fo-DTS). This applies also to lines 513-515. TIR is a method for pattern identification. However, you need to describe this already before and not only in Advantages and Limitations. See also comment regarding this topic below.

L443ff: Furthermore, you should briefly mention typical approaches to measure temperature and in this paragraph you should also include fibre-optic distributed temperature sensing even if it has not been used in braided rivers yet.

L443ff: You could also consider adding temperature methods that don't rely on natural

temperature differences but use temperature as an active tracer, e.g. active (heated) DTS, heat-pulse sensors etc.

L524: "Hydraulic property measurements" is no suitable chapter headline for the sub-chapter "Groundwater observation wells"! Alternatives might be "2.4 Flow-net analysis" or "2.4 Darcy approach". I would call 2.4.1 "Hydraulic gradients" and 2.4.2 "Hydraulic conductivity".

L525ff: The second sentence of the paragraph is wrong: The groundwater level/hydraulic gradient is no hydraulic property. Hydraulic properties are the hydraulic conductivity, the porosity etc. The rest of the paragraph belongs to 2.4.2.

L559ff: You use the terms well, piezometer and mini-piezometer but I have not seen a definition of those terms. Consider to include also other designs, e.g. M. O. Rivett, R. Ellis, R. B. Greswell, R. S. Ward, R. S. Roche, M. G. Cleverly, C. Walker, D. Conran, P. J. Fitzgerald, T. Willcox, and J. Dowle. Cost-effective mini drive-point piezometers and multilevel samplers for monitoring the hyporheic zone. Quarterly Journal of Engineering Geology and Hydrogeology 41:49-60, 2008. However, in this paragraph with its focus on groundwater level measurements either sufficient diameter for a logger or an electric contact gauge is useful even though some scientists used innovative approaches for very small diameters (transparent tubes, suction to increase water level differences to an easily visible height, colored strings ...) Also, you should consider describing at least in brief typical installation techniques for the different designs and different depth depending on substrate quality. Furthermore, report at least in one sentence how water tables are measured/logged.

L605ff: Consider to add also in brief the use of geophysics to characterize the subsurface pattern (together with some core for calibration of geophysical methods).

L642ff: Mention that loggers require a certain diameter of wells/piezometers as a further disadvantage.

Table 1: You have split the first method (water budget) into two budget methods. Why haven't you also split the following methods as in the text (e.g. environmental tracers, heat tracers, ...). In fact heat tracers are also an environmental tracer. Why are River reach budgets suitable only for relatively homogenous aquifers? Remove pH and DO from environmental tracers (see corresponding comments above). As far as I understand the table and its table capitations it is about methods for quantifying water fluxes. The point "Aerial surveys can be faster than in-stream surveys" does not fit. This is a method for pattern identification and not for flux determination. As described above I doubt that "Hydraulic Property Measurement" is an adequate headline for this type of method. I don't think that this applies only to minipiezometers. Piezometers are also easy and quick to install.

In general other authors have grouped their methods into three categories and I think this would be advantageous here as well:

+ point methods to estimate fluxes at a discrete location

+ methods for pattern identification don't yield numbers for fluxes but can help to identify representative sites and the most extreme sites to conduct the point methods at the most interesting sites. Under certain circumstances also transfer functions possible that combine methods for pattern identification and point methods

+ integrating methods over large areas that result in total fluxes, but without any information about local fluxes or distribution of patterns

L783ff: Please keep the three points above in mind. Remote sensing is not gathering the same information as the point methods mentioned in L781-783! The same applies to Line 870-872.

L797f: Please mention here also that time series that might be recorded with loggers can be very useful to gain system understanding because groundwater-surface water interactions might vary with time and even the flow direction might reverse over time.

L849: It is definitely strange to have a subchapter 3.1 but no 3.2. Also, it is confusing that the introduction before 3.1 is about 5 pages long and 3.1 less than 1 page long.

MINOR COMMENTS:

================

L48: Cite also Winter et al. (1998) (https://pubs.usgs.gov/circ/circ1139/)

L57: .

L102f: Why is improved knowledge of historial patterns needed? In addition, can you please cite a reference.

L118: A more scientific reference would be great here.

L147:         Consider     adding     Rosenberry     et     al.     (2015) (https://onlinelibrary.wiley.com/doi/full/10.1002/hyp.10403)

L279: I think what is much more important than evenly distributed groundwater discharge or recharge is an even groundwater concentration.

L289ff: Please correct: there are three stable oxygen isotopes including O-17!

L291f: "The process is largely driven by temperature, whereby . . . at higher elevation due to colder temperatures" The process is not driven by elevation but the elevation effect is a result of decreasing temperatures with increasing depth. In case you really want to mention processes in addition to temperature you can add humidity and salinity as further processes.

L519: I think the most important point that should be measured here is season!

L560: Only deep wells/piezometers are expensive.

L565: Isn't this also a conceptual diagram of a well?

L656: You might want to mention that it is nearly impossible to take undisturbed

cores/rings for KSat analysis if the sediment contains coarse gravel as this is the case in most braided streams. L674: "interactions" instead of "interaction"

L687: Delete: "and will not be repeated here."

L704: "They used" instead of "The used"

L754: You are not investigating groundwater and surface water but their interactions: "... for investigaton of groundwater-surface water interactions, and there ..."

L764: "a study" instead of "the study"

L808: "by the study objective and the study object"

L820: Only during storms???

L851: "One of the most ..." – I do not understand this sentence.

L854: Consider adding here S. Krause, D. M. Hannah, J. H. Fleckenstein, C. M. Heppell, D. Kaeser, R. Pickup, G. Pinay, A. L. Robertson, and P. J. Wood. Interdisciplinary perspectives on processes in the hyporheic zone. Ecohydrology 4 (4):481-499, 2011.

L869: the present paper

L895: You might add here DTS and geophysics

---

## Referee Comment (RC2) · Anonymous Referee #2 · 15 Feb 2019

This paper provides a thorough literature review of papers describing methods to characterize groundwater-surface water (gw-sw) interaction in streams. The emphasis of the paper are studies in braided streams. Braided streams provide additional challenges for characterization, including ephemeral and low flows, heterogeneity, invasive species, and anthropogenic alteration as discussed in the paper.

The paper points out that relatively fewer studies of gw-sw exchange have been conducted in braided streams. This review paper pulls together a summary of many such studies. However, the paper lacks synthesis on several points. These issues should be addressed to strengthen the paper:

[Figure]

1. While the paper points out challenges in measuring gw-sw interaction, including a nice summary table by method, the challenges do not link up with the specific issues in braided rivers. In other words, the challenges would apply to all river types. For instance, the authors mention how heterogeneity makes it difficult to measure flow. While braided streams may be more heterogeneous (however that is defined), all streams would benefit from methods that address heterogeneity. I would have liked to see how the cross-sectional heterogeneity (in contrast to along reach) impacts measurement techniques. That said, Genereux's group has some papers illustrating both along-reach and cross section variation in streambed K, so even this aspect is not unique to braided rivers. The advantages and disadvantages sections list challenges that would apply to other stream types as well. The abstract and conclusions emphasize the need for multiple methods and consideration of scale in selecting methods, but again these recommendations apply to any river type. Without details about why a particular method works elsewhere but not on braided rivers, the paper lacks focus. It does not suffice to say a method is "more difficult" when it is difficult in a variety of river settings.

2. The paper provides a map of locations with braided streams, but does not justify why these locations are included and not others. The definition of what "concentrated" means in terms of distribution of braided streams is not provided. There is a list of braided streams in the US on https://commons.wikimedia.org/wiki/Category:Braided_rivers_in_the_United_States_by_state, which suggested that braided rivers are important in the US too, yet no sites there are listed. To list the map as a significant feature of the paper ("to the authors' knowledge, this is the first map of its kind") but provide no details on how the map was generated is frustrating to the reader.

3. The word "hyporheic" only appears in the abstract and end of the paper, not in the main body. This mention in the abstract should be removed since it is not a topic covered in the paper. It is probably better left to another paper as the issues in measuring

hyporheic flow differ significantly.

4. The modeling discussion is focused too much on MODFLOW. The description of MODFLOW packages can be found elsewhere and there are other models that incorporate groundwater-surface water interaction that could be discussed. For example, a recent special issue in Groundwater on integrated modeling included a paper on streambed heterogeneity. There is also a recent review paper on modeling gw-sw interaction in Reviews of Geophysics that provides a broader view. The abstract mentions the need for new approaches in modeling, but the paper does not provide sufficient direction to justify this as a conclusion of the paper. The conclusion the models need more data and more sensitivity analysis has been stated many times before.

5. I was surprised that fiber optic temperature systems (also known as DTS for distributed temperature systems) and geophysics were not discussed. These methods have been mentioned in other reviews and provide broader coverage which might benefit braided streams. I found it odd to bring up thermal imaging for the first time in the discussion section rather than in the review of methods, especially since it is mentioned in the abstract and it is one of the more promising techniques for heterogeneous systems. An example of the benefits of thermal imaging might provide an interesting figure.

6. On the topic of figures, the figures were lacking in illustrative examples of applications. There was a map, but the other figures were photos or diagrams and didn't show quantitative challenges or opportunities. In other words, I think it would help the readers' understanding to include data figures.

7. One place that the paper focuses on braided streams is the literature review of methods. The paper summarizes applications in braided streams and the table of methods lists braided stream citations. However, the literature summary sections of the paper are a bit dry. They list highlights of each paper one after another. I think some of these papers could be describing non-braided streams and the reader would

not know. This type of literature review needs to be briefer and provide synthesis of issues specific to the problem identified. In addition, a significant number of references (estimated 25% based on the first page of the bibliography) are not readily available literature but reports or theses (typically from NZ). Many readers will not have ready access and the focus on one region is not justified.

It can be difficult to meet the standards of a review article. In the end, I ask myself whether I would give this paper to colleagues to read, or just keep recommending Kalbus et al. or LaBaugh and Rosenberry as review papers on the topic. I do not think there is enough new material here for me to consider this paper to be an update on the earlier papers. If revising, I would recommend a very short review paper, which introduces Table 1 and gives the reader the reference list for readers to select topics on their own (rather than the one line summaries of each paper). The shorter paper also needs to provide the reader with an approach to braided streams that is distinctly different than other streams – this message will take additional synthesis and thus I would consider it to be a new paper rather than a resubmission. Hence, I am recommending rejection and significant redirection for any new submittal.

Some papers mentioned in the review:

Berg, S.J., Grosso, N.R., Sherrier, M.P., Mudrick, K., Ohr, M., Hwang, H.T., Park, Y.J., Callaghan, M.V., Frey, S.K. and Sudicky, E.A., 2019. Natural stimuli calibration with fining direction regularization in an integrated hydrologic model. Groundwater, 57(1), pp.21-35.

Genereux, D.P., Leahy, S., Mitasova, H., Kennedy, C.D. and Corbett, D.R., 2008. Spatial and temporal variability of streambed hydraulic conductivity in West Bear Creek, North Carolina, USA. Journal of Hydrology, 358(3-4), pp.332-353.

Brunner, P., Therrien, R., Renard, P., Simmons, C.T. and Franssen, H.J.H., 2017. Advances in understanding river‐groundwater interactions. Reviews of Geophysics, 55(3), pp.818-854.

Not mentioned, but a classic that should be cited: Winter, TC, Harvey, JW, Franke, OL and Alley, WM. 1998 Ground water and surface water; a single resource. USGS Circular 1139
* * *

---

## Author Comment (AC1) · 18 Mar 2019

**Reply to Reviewer # 1**

**General Comments**

*1.      The manuscript "A review of methods for measuring groundwater-surface water exchange in braided rivers" by Katie Coluccio and Leanne Kaye Morgan is a review paper. As the title suggests it is about measuring methods for groundwater-surface water exchange in braided rivers. In general, the manuscript is informative, provides an overview about the current literature, is well structured and well written.*

Response: We thank the reviewer for these positive comments.

*2.      However, some sections are lengthy and might be shortened.*

Response: We will shorten the revised manuscript, as discussed below.

*3.      Furthermore, as indicated in the major comments below important information, definitions etc. is missing.*

Response: We thank the review for pointing out these omissions. As detailed below, we will address this in the revised manuscript.

*4.      In general the authors could think a little bit more outside of the box. They are very focused on the methods that have been used in studies of groundwater-surface water interactions in braided rivers. But there are several similar groundwater-surface water interfaces and as part of a scientific paper I would expect the authors to consider additional methods that might be adapted to braided rivers in future in addition to simply summarising the literature available at present.*

Response: As detailed below, in the revised manuscript we will discuss additional methods that have not yet been applied in braided rivers but show potential such as fibre-optic temperature sensing, active heat pulse methods, artificial tracers, remote collection of data via satellite imaging, and geophysical techniques.

*5.      I think the manuscript can be published after revision.*

Response: Thank you.

**Major comments**

*6.      Entire manuscript: Try to shorten your manuscript and avoid lengthy descriptions of the literature, e.g. L173-L213, L216-242, L289-325, L328-L379, L382-402, L533-603, L606-L640, L667-L739.*

Response: We agree that the manuscript would benefit from more concise descriptions of the literature. We will adjust this in the revised draft.

*7.      L60 & entire manuscript: Suggest also methods that have been successfully used at other groundwater-surface water interfaces and that might be adapted to braided rivers and might be used in braided rivers in future. Reporting only what has already been done in braided rivers is a little bit thin.*

Response: Thank you for the suggestion. In the revised manuscript we will add a discussion of techniques used in other environments that might be useful in braided rivers, including geophysical methods, additional temperature methods (DTS, active heat pulse methods), more remote sensing techniques (satellite imagery), and artificial tracers.

*8.      L64 & Fig. 1 & L882: I strongly recommend adding all additional instances of braided rivers outside of the major regions. You might use different symbols for major regions with braided rivers and single instances.*

Response: When initially creating Fig. 1, we had considered attempting to include all instances of braided rivers globally, as suggested. However, we decided against this for a few reasons. Mainly, we were concerned that stating we had accounted for "all" braided rivers would run the risk of missing some rivers and in so doing being factually incorrect. Secondly, we felt that highlighting the locations where most braided rivers occur would be most useful to readers, as this indicates where most of the braided rivers research has been conducted. In an attempt to account for instances of braided rivers outside of the major regions, we intend to add a sentence or two at L64 noting that braided rivers also occur in small numbers in the U.S., Scotland, Iceland, China, Poland, Belarus, Colombia, Congo, Brazil, Paraguay, Argentina, and the Touat Valley in Africa. Also, we will add Russia to Fig. 1 based on

comments in studies by Chalov & Alexeevsky (2015) and Alexeevsky et al. (2013) about the high number of braided rivers in that country.

9.  *L100f; L791f, L855: I think there is a need for clear definitions of "groundwater-surface water interactions" and of "hyporheic exchange". Often, the term "groundwater-surface water interaction" is used in literature in a wide sense including hyporheic exchange as one process of groundwater-surface water interactions. However, according to line 100f you consider both as separate processes with some impacts on each other.*

Response: Thank you for highlighting this, and we agree that this is an area where more clarity would be helpful. We propose adding text similar to the below in the revised manuscript: "This paper often refers to groundwater-surface water exchange, which in this context may include regional groundwater exchange with river water, as well as hyporheic zone exchange. Hyporheic exchange refers to downwelling or upwelling of water through the hyporheic zone, i.e., the saturated area between the streambed and shallow aquifer where stream water and shallow groundwater mix."

10.  *L134ff: Even though I agree that there is little research about groundwater-surface water interactions in braided rivers your "Web of Science" search is meaningless. I tried to reproduce it. First of all "groundwater and surface water interactions" with "…" results in much smaller numbers than the ones reported by you, e.g. only three papers for lakes instead of 437 reported by you. Repeating the search without "…" resulted in approximately the numbers reported by you. However, having a closer look at those papers revealed that most of the hits are not about groundwater-surface water interactions at all but that the separate words of the phrase are used in separate sentences and in different context. Furthermore, at many of the interfaces mentioned by you (lakes, ocean, stream) specific terms are used, e.g. "lacustrine groundwater discharge", "submarine groundwater discharge" and "hyporheic zone" instead of "groundwater and surface water interactions". Sometimes the word "interactions" is substituted by "exchange" or by "interfaces". Also, there are different spellings for "groundwater" such as "ground water". I am quite sure that the largest number of studies focusing on groundwater-surface water interactions is about stream, followed by (coastal) oceans followed by lakes and finally by braided rivers. You might also have a look at review papers focusing on the different interfaces. There are several of them. I*

*recommend either deleting lines 134-139 or repeating this literature search with a set of different keywords to get a more comprehensive overview of the literature of interest.*

Response: Thank you for highlighting the issues with L134-139, and we agree that deleting these lines would improve the manuscript.

*11.    L158ff: From my experience budgets are often quite error-prone because accurate measurements of river discharge are challenging. Often changes in river discharge between stations are much smaller than the error inherent to the measurements. You should mention this shortcoming more clearly than only in lines 261-263.*

Response: Indeed, this is an important factor to consider. Additional to L261-263, we have mentioned this limitation in Table 1 (under River Reach Water Budgets). We will also add a comment about this in the discussion of the revised manuscript.

*12.    L272ff/L284ff: I think it is important to introduce here also the concept that tracers need to be conservative (on the scale of the investigation). In this context, I doubt that dissolved oxygen (L284), nitrate (L285), sulphate (L286) and pH (L404) are useful tracers. pH might be acceptable in the context of alkanity but that also needs more discussion. The concentrations of oxygen, nitrate, sulfate and H+ will be altered due to many different biogeochemical processes. They might be used under certain circumstances and on small scales on which little turnover takes place. But this is something very critical. If you list these compounds you need to discuss them critically.*

Response: Thank you for the comments here and we agree with your point that tracers need to be conservative, and this is an important consideration to make when selecting parameters to measure. Where dissolved oxygen, nitrate, sulphate and pH have been discussed, in the revised manuscript we will add comments on the limitations of these parameters. We think that these parameters are still worthy of discussion as they have been used in several previous studies to varying degrees of success.

*13.    L272ff: In addition to environmental tracers I recommend to discuss also artificial tracers that might be added to the system. There are multiple studies using artificial tracers and I am quite sure that the also have been used in braided rivers. However, even if not they are an option that should be considered.*

Response: Indeed, we also suspect that there have been several studies using artificial tracers in braided rivers, however we have been unable to find published research other than that of Dann et al. (2008) who used dye tracers to characterise a braided river-deposited aquifer in New Zealand. We will include that study in the revised manuscript. We will also discuss some artificial tracer studies (e.g., Langston et al. (2013), Ferreira et al. (2018), Flury & Wai (2003)) conducted in other environments that may be useful to readers.

*14. L457-468: I don't see any connection of this paragraph to the topic groundwater-surface water interactions. Therefore, I recommend deleting this paragraph.*

Response: Thank you for highlighting this, and we agree that this study was not specifically related to investigating groundwater-surface water interactions, and thus should be removed. However, given this is an example of thermal infrared imaging used in a braided river setting, we will include it as a proposed method for highlighting temperature gradients in riverbeds as an indicator of groundwater-surface water exchange.

*15. L469-484: The topic of the present review is measurement methods for groundwater-surface water interactions. Thus, these two paragraphs don't fit to the topic of the review paper. They are about impacts of groundwater and surface water on temperature (and ecological consequences) but not how to use measurements to identify groundwater-surface water interactions.*

Response: Thank you for the constructive comments on these studies. We will delete these two paragraphs. The two studies mentioned in these paragraphs (i.e., Acuna and Tockner, 2009; Malard et al., 2001) used multiple methods to assess groundwater-surface water exchange, and thus we feel that these are useful references to include, albeit only in Table 1 within the revised manuscript.

*16. L502ff: I think it is important to measure temperature depth profiles as you do in this paragraph. However, you should go into a little bit more detail here and also mention typical evaluation methods for temperature depth profiles such as the steady state approach (e.g. C. Schmidt, M. Bayer-Raich, and M. Schirmer. Characterization of spatial heterogeneity of groundwater-stream water interactions using multiple depth streambed temperature*

*measurements at the reach scale. Hydrology and Earth System Sciences 10:849-859, 2006)*
*or VFLUX.*

Response: We agree that it would be useful to include some more detail on how temperature
depth profiles may be collected and analysed. We will amend this in the revised manuscript.

*17.     L443ff: I think at one point in this subchapter you should clearly differentiate between
methods that are used to determine fluxes (e.g. temperature depth profiles) and methods for
pattern identification (aerial TIR, fo-DTS). This applies also to lines 513-515. TIR is a
method for pattern identification. However, you need to describe this already before and not
only in Advantages and Limitations. See also comment regarding this topic below.*

Response: This is a very good suggestion, thank you. At the beginning of Section 2.3, we will
clarify the difference between temperature methods to detect patterns and those used to
measure fluxes. We agree that it is important to note the differences in the methods, and we
have referenced papers that use methods in both categories. We will also discuss this in more
detail in Section 2.3.1 under Advantages/Limitations of heat tracers.

*18.     L443ff: Furthermore, you should briefly mention typical approaches to measure
temperature and in this paragraph you should also include fibre-optic distributed
temperature sensing even if it has not been used in braided rivers yet.*

Response: We agree that it would be useful to have a brief explanation of typical approaches
to measuring temperature while noting which ones are used for pattern recognition or flux
estimates (as per the comment above). We will discuss fibre-optics in the "Key Gaps and
Possibilities" section in the revised manuscript (citing studies in other environments such as
Lovett et al. (2015), Meijer (2015), Briggs et al. (2014), Rosenberry et al. (2016), Busato et
al. (2019), Klinkenberg (2015)).

*19.     L443ff: You could also consider adding temperature methods that don't rely on
natural temperature differences but use temperature as an active tracer, e.g. active (heated)
DTS, heat-pulse sensors etc.*

Response: Thank you for the suggestion, and we agree that it would be useful to include
active heat tracers (such as the 3D heat pulse array used in Banks et al. (2018)) in the revised

manuscript. To our knowledge, these methods have not yet been used in a braided river setting, but they do have potential and thus may be beneficial for readers. We will include these methods in the "Key Gaps and Possibilities" section of the revised manuscript.

*20.      L524: "Hydraulic property measurements" is no suitable chapter headline for the subchapter "Groundwater observation wells"! Alternatives might be "2.4 Flow-net analysis" or "2.4 Darcy approach". I would call 2.4.1 "Hydraulic gradients" and 2.4.2 "Hydraulic conductivity".*
Response: Thank you for the suggestions and we propose to amend the section headings to the following:

2.4 Darcy approach

2.4.1 Hydraulic gradient

2.4.2 Hydraulic conductivity

Advantages and Limitations

*21.      L525ff: The second sentence of the paragraph is wrong: The groundwater level/hydraulic gradient is no hydraulic property. Hydraulic properties are the hydraulic conductivity, the porosity etc. The rest of the paragraph belongs to 2.4.2.*
Response: Thank you for highlighting this. We will delete lines 525 to 530 as these points are covered in sections 2.4.1 and 2.4.2. This will also serve to shorten the manuscript.

*22.      L559ff: You use the terms well, piezometer and mini-piezometer but I have not seen a definition of those terms. Consider to include also other designs, e.g. M. O. Rivett, R. Ellis, R. B. Greswell, R. S. Ward, R. S. Roche, M. G. Cleverly, C. Walker, D. Conran, P. J. Fitzgerald, T. Willcox, and J. Dowle. Cost-effective mini drive-point piezometers and multilevel samplers for monitoring the hyporheic zone. Quarterly Journal of Engineering Geology and Hydrogeology 41:49-60, 2008. However, in this paragraph with its focus on groundwater level measurements either sufficient diameter for a logger or an electric contact gauge is useful even though some scientists used innovative approaches for very small diameters (transparent tubes, suction to increase water level differences to an easily visible height,*

*colored strings ...) Also, you should consider describing at least in brief typical installation techniques for the different designs and different depth depending on substrate quality. Furthermore, report at least in one sentence how water tables are measured/logged.*

Response: In L559 we intended "groundwater well" and "piezometer" to be synonymous. To clarify this, we will modify the sentence to read: "In terms of specific methods that can be used for measurements, existing piezometers (i.e., monitoring wells) near rivers can be useful for conducting these types of studies, particularly given the high cost of drilling new wells." At L561 we will add the sentence: "Please refer to standard text such as Fetter (2001) for a definition of piezometers". In L561 we have defined "mini-piezometers" as "scaled-down versions of piezometers and typically installed no deeper than about two metres". With respect we prefer not to include reference to installation methods as these are detailed in the cited references. Also, for the sake of brevity, we prefer not to detail other piezometer designs.

In the revised manuscript, we will comment in section 2.4.3 about the need to consider the diameter of wells being used with downhole equipment (e.g., loggers). Also, at the beginning of section 2.4.1 we will briefly detail the way in which water levels are measured.

*23. L605ff: Consider to add also in brief the use of geophysics to characterize the subsurface pattern (together with some core for calibration of geophysical methods).*

Response: This is a good suggestion, thank you. In the revised manuscript, we will discuss new methods for use in braided rivers in the "Key gaps and possibilities section". For the sake of brevity, we will discuss geophysics in that section.

*24. L642ff: Mention that loggers require a certain diameter of wells/piezometers as a further disadvantage.*

Response: We will add this to the revised manuscript.

*25. Table 1: You have split the first method (water budget) into two budget methods. Why haven't you also split the following methods as in the text (e.g. environmental tracers, heat tracers, ...). In fact heat tracers are also an environmental tracer. Why are River reach budgets suitable only for relatively homogenous aquifers? Remove pH and DO from*

*environmental tracers (see corresponding comments above). As far as I understand the table and its table capitations it is about methods for quantifying water fluxes. The point "Aerial surveys can be faster than in-stream surveys" does not fit. This is a method for pattern identification and not for flux determination. As described above I doubt that "Hydraulic Property Measurement" is an adequate headline for this type of method. I don't think that this applies only to minipiezometers. Piezometers are also easy and quick to install. In general other authors have grouped their methods into three categories and I think this would be advantageous here as well:*

*+ point methods to estimate fluxes at a discrete location*

*+ methods for pattern identification don't yield numbers for fluxes but can help to identify representative sites and the most extreme sites to conduct the point methods at the most interesting sites. Under certain circumstances also transfer functions possible that combine methods for pattern identification and point methods*

*+ integrating methods over large areas that result in total fluxes, but without any information about local fluxes or distribution of patterns.*

Response: Thank you for your thorough comments on Table 1. The intention of this table was to summarise all the methods discussed in the review, both for identifying patterns and for estimating fluxes. Perhaps the table title has created the confusion here, so we will amend the title to read "Advantages and disadvantages of various methodologies for **measuring** groundwater-surface water interactions in braided rivers".

We are not convinced that organising the methods according to scale of measurement would be helpful as there would be overlap amongst methods (i.e., some methods could be used at multiple scales, see Fig. 1 in Kalbus et al. (2006)). The proposed revised categories for Table 1 are: Water budgets, Hydrochemistry, Temperature studies, Darcy approach, Modelling, Artificial tracers, Geophysics and Remote sensing.

*26.    L783ff: Please keep the three points above in mind. Remote sensing is not gathering the same information as the point methods mentioned in L781-783! The same applies to Line 870-872.*

Response: Thank you for highlighting that we may need more clarity around scales of measurement. However, we are not sure why there is confusion here. Depending on how they are carried out, the methods mentioned in L781-783 (pumping tests, flow gauging, stable

isotope analysis, solute tracers and chemical analysis) can provide broad spatial scale information, as can TIR imaging, geophysical methods and satellite data. As we mentioned in the response to the comment on Table 1 above, many of these methods can be used to collect data at various scales, while some indeed are point methods only (e.g. permeameter tests or 1-D temperature profiles).

*27.     L797f: Please mention here also that time series that might be recorded with loggers can be very useful to gain system understanding because groundwater-surface water interactions might vary with time and even the flow direction might reverse over time.*
Response: Indeed, this is an important point to make. In this section of the manuscript, we attempted to illustrate this with the example of temperature time series data in L801-803, but in the revised manuscript we will make it clearer that time series data for a range of parameters can be very useful to collect to observe changes in groundwater-surface water interactions over time.

*28.     L849: It is definitely strange to have a subchapter 3.1 but no 3.2. Also, it is confusing that the introduction before 3.1 is about 5 pages long and 3.1 less than 1 page long.*

Response: Thank you for highlighting this. We propose to change the numbering of Section 3.1 (Key gaps and possibilities) to Section 4.

**MINOR COMMENTS**

*29.     L48: Cite also Winter et al. (1998) (https://pubs.usgs.gov/circ/circ1139/)*

Response: Thank you for the relevant suggestion. This reference will be added in the revised manuscript.

*30.     L102f: Why is improved knowledge of historical patterns needed? In addition, can you please cite a reference.*
Response: Better knowledge of historic states and patterns of braided rivers would be very helpful for understanding the implications of modifications to natural systems in order to set

water allocation limits and minimum flow levels in rivers (Reigler, 2012; Burbery et al., 2010). For example, many irrigation schemes have artificially raised groundwater levels due to land surface recharge, or lowered groundwater levels due to abstraction in comparison to their natural (pre-irrigation) states. In some rivers this has affected the losing/gaining patterns.

*31.     L118: A more scientific reference would be great here.*

Response: The reference in the previous draft will be replaced by references to Caruso (2006); Larned et al. (2008); Tockner and Stanford (2002), which are all peer-reviewed publications in international journals.

*32.     L147: Consider adding Rosenberry et al. (2015)*
*(https://onlinelibrary.wiley.com/doi/full/10.1002/hyp.10403)*
Response: This is a very useful reference, and we will add it to the revised manuscript, along with Brunner et al. 2017, which is a useful review of the latest advances in methods for characterising and modelling river and groundwater interactions and specifically mentions braided streams in some parts.

*33.     L279: I think what is much more important than evenly distributed groundwater discharge or recharge is an even groundwater concentration.*
Response: Thank you for the suggestion, and we agree that amending the wording to "an even groundwater concentration" would be more accurate.

*34.     L289ff: Please correct: there are three stable oxygen isotopes including O-17!*

Response: Thank you for highlighting this oversight. Oxygen-17 will be added to this discussion of stable oxygen isotopes.

*35.     L291f: "The process is largely driven by temperature, whereby ... at higher elevation due to colder temperatures" The process is not driven by elevation but the elevation effect is*

*a result of decreasing temperatures with increasing depth. In case you really want to mention processes in addition to temperature you can add humidity and salinity as further processes.*

Response: Thanks for pointing this out. We will amend this sentence to read: "The process is largely driven by temperature, whereby precipitation is increasingly depleted in $^{18}$O at colder temperatures (which tend to occur at higher elevations) (Sharp, 2007)".

*36.    L519: I think the most important point that should be measured here is season!*

Response: Thank you, this will be added to the revised manuscript.

*37.    L560: Only deep wells/piezometers are expensive.*

Response: Agreed. We will amend the text to reflect that cost of installing wells or piezometers may only be prohibitively high in some situations.

*38.    L565: Isn't this also a conceptual diagram of a well?*

Response: As detailed in our response to point 22 above, we will modify the text at L559 so that it is clear that we consider piezometers to be monitoring wells. We will also make it clear that mini-piezometers are small versions of piezometers. In light of this we think that the labelling of this figure as a conceptual diagram of a mini-piezometer is no longer confusing.

*39.    L656: You might want to mention that it is nearly impossible to take undisturbed cores/rings for KSat analysis if the sediment contains coarse gravel as this is the case in most braided streams.*

Response: Thank you. We will add a sentence to this effect.

*40.    L 674: "interactions" instead of "interaction"*

Response: Thank you. The text will be amended.

*41.    L687: Delete: "and will not be repeated here."*

Response: Thank you. This text will be deleted.

*42.    L704: "They used" instead of "The used"*

Response: Thank you. This suggested change will be made.

*43.    L754: You are not investigating groundwater and surface water but their*
*interactions:*
*"... for investigation of groundwater-surface water interactions, and there ..."*

Response: Thank you for recognising this error. The text will be amended to reflect your comment.

*44.    L764: "a study" instead of "the study"*

Response: Thank you. The suggested change will be made.

*45.    L808: "by the study objective and the study object*

Response: Thank you. The suggested text will be added.

*46.    L820: Only during storms???*

Response: This is a fair point. Our intention in specifically mentioning storms was to highlight mass sediment movement during flood events, but indeed, sediment transport at other times may equally damage equipment. The mention of storms here will be removed.

*47.    L851: "One of the most ..." – I do not understand this sentence.*

Response: In the revised manuscript we will amend L851-853 to the following: "One of the most significant gaps in this area relates to how hyporheic flow processes operate and how they impact river flow levels and water quality in braided rivers."

*48. L854: Consider adding here S. Krause, D. M. Hannah, J. H. Fleckenstein, C. M. Heppell, D. Kaeser, R. Pickup, G. Pinay, A. L. Robertson, and P. J. Wood. Interdisciplinary perspectives on processes in the hyporheic zone. Ecohydrology 4 (4):481-499, 2011.*

Response: This is an excellent suggested reference for this line, as well as to enhance the discussion of the hyporheic zone in the present paper. This reference will be added here.

*49.     L869: the present paper*

Response: This suggested text will be added.

*50.     L895: You might add here DTS and geophysics*

Response: We agree that it would be helpful to add DTS and geophysics to this line.

**References**

Acuña, V., & Tockner, K. (2009). Surface-subsurface water exchange rates along alluvial river reaches control the thermal patterns in an Alpine river network. *Freshwater Biology, 54,* 306–320. doi: 10.1111/j.1365-2427.2008.02109.x

Alexeevsky, N. I., Chalov, R. S., Berkovich, K. M., & Chalov, S. R. (2013). Channel changes in largest Russian rivers: Natural and anthropogenic effects. *International Journal of River Basin Management, 11*(2), 175-191. doi: 10.1080/15715124.2013.814660

Banks, E. W., Shanafield, M. A., Noorduijn, S., McCallum, J., Lewandowski, J., & Batelaan, O. (2018). Active heat pulse sensing of 3-D-flow fields in streambeds. *Hydrology and Earth System Sciences, 22*, 1917–1929. doi: 10.5194/hess-22-1917-2018

Briggs, M. A., Lautz, L. K., Buckley, S. F., & Lane, J. W. (2014). Practical limitations on the use of diurnal temperature signals to quantify groundwater upwelling. *Journal of Hydrology, 519*, 1739-1751. doi: 10.1016/j.jhydrol.2014.09.030

Brunner, P., Therrien, R., Renard, P., Simmons, C. T., & Franssen, H.-J. H. (2017). Advances in understanding river-groundwater interactions. *Reviews of Geophysics, 55*, 818–854. doi: 10.1002/2017RG000556

Burbery, L., & Ritson, J. (2010). *Integrated study of surface water and shallow groundwater resources of the Orari catchment*. Environment Canterbury Report Number R10/36. Christchurch, New Zealand. Retrieved from http://docs.niwa.co.nz/library/public/ECtrR10-36.pdf

Busato, L., Boaga, J., Perri, M. T., Majone, B., Bellin, A., & Cassiani, G. (2019). Hydrogeophysical characterization and monitoring of the hyporheic and riparian zones: The Vermigliana Creek case study. *Science of the Total Environment, 648*, 1105–1120. doi: 10.1016/j.scitotenv.2018.08.179

Caruso, B. S. (2006). Project River Recovery: Restoration of Braided Gravel-Bed River Habitat in New Zealand's High Country. *Environmental Management, 37*(6), 840-861. doi: 10.1007/s00267-005-3103-9

Chalov, S. R., & Alexeevsky, N. I. (2015). Braided rivers: Structure, types and hydrological effects. *Hydrology Research, 46*(2), 258-275. doi: 10.2166/nh.2013.023

Dann, R. L., Close, M. E., Pang, L., Flintoft, M. J., & Hector, R. P. (2008). Complementary use of tracer and pumping tests to characterize a heterogeneous channelized aquifer system in New Zealand. *Hydrogeology Journal, 16*, 1177–1191. doi: 10.1007/s10040-008-0291-4

Ferreira, V. V. M., Moreira, R. M., Rocha, Z., Chagas, C. J., Fonseca, R. L. M., Santos, T. O., . . . Menezes, M. A. B. C. (2018). Use of radon isotopes, gamma radiation and dye tracers to study water interactions in a small stream in Brazil. *Environmental Earth Sciences, 77*(19), 1-12. doi: 10.1007/s12665-018-7879-3

Fetter, C. W. (2001). *Applied Hydrogeology* (4th ed.): Pearson Education.

Flury, M., & Wai, N. N. (2003). Dyes as tracers for vadose zone hydrology. *Reviews of Geophysics, 41*(1), 1002-1037. doi: 10.1029/2001RG000109

Langston, G., Hayashi, M., & Roy, J. W. (2013). Quantifying groundwater-surface water interactions in a proglacial moraine using heat and solute tracers. *Water Resources Research, 49*, 5411–5426. doi: 10.1002/wrcr.20372

Larned, S. T., Hicks, D. M., Schmidt, J., Davey, A. J. H., Dey, K., Scarsbrook, M., . . . Woods, R. A. (2008). The Selwyn River of New Zealand: A benchmark system for

alluvial plain rivers. *River Research and Applications, 24*(1), 1-21. doi: 10.1002/rra.1054

Lovett, A., Cameron, S., Reeves, R., Meijer, E., Verhagen, F., van der Raaij, R., . . . Morgenstern, U. (2015). *Characterisation of groundwater-surface water interaction at three case study sites within the Upper Waikato River Catchment using temperature sensing and hydrochemistry techniques* (GNS Science Report 2014/64). Retrieved from http://shop.gns.cri.nz/sr_2014-064-pdf/

Kalbus, E., Reinstorf, F., & Schirmer, M. (2006). Measuring methods for groundwater–surface water interactions: A review. *Hydrology and Earth System Sciences, 10*, 873–887.

Klinkenberg, J. (2015). *Characterising groundwater-surface water interaction using fibre-optic distributed temperature sensing and validating techniques in Whakaipo Bay, Lake Taupo, New Zealand.* Masters Thesis. Utrecht University. Retrieved from https://dspace.library.uu.nl/handle/1874/324367

S. Krause, D. M. Hannah, J. H. Fleckenstein, C. M. Heppell, D. Kaeser, R. Pickup, G. Pinay, A. L. Robertson, and P. J. Wood. Interdisciplinary perspectives on processes in the hyporheic zone. *Ecohydrology, 4* (4), 481-499, 2011.

Malard, F., Mangin, A., Uehlinger, U., & Ward, J. V. (2001). Thermal heterogeneity in the hyporheic zone of a glacial floodplain. *Canadian Journal of Fisheries and Aquatic Sciences, 58*(7), 1319–1335. doi: 10.1139/cjfas-58-7-1319

Meijer, E. C. (2015). *Using fibre-optic distributed temperature sensing and heat modelling to characterize groundwater- surface water interaction in Whakaipo Bay, Lake Taupo, New Zealand.* Utrecht University, Utrecht, The Netherlands. Retrieved from https://dspace.library.uu.nl/handle/1874/311429

Riegler, A. (2012). *Influence of groundwater levels on zero river flow: North Branch, Ashburton River, New Zealand.* University of Vienna, Austria. Retrieved from http://othes.univie.ac.at/22451/1/2012-06-17_0600876.pdf

Rosenberry, D. O., Lewandowski, J., Meinikmann, K., & Nützmann, G. (2015). Groundwater - the disregarded component in lake water and nutrient budgets. Part 1: effects of groundwater on hydrology. *Hydrological Processes, 29*, 2895–2921. doi: 10.1002/hyp.10403

Rosenberry, D. O., Briggs, M. A., Delin, G., & Hare, D. K. (2016). Combined use of thermal methods and seepage meters to efficiently locate, quantify, and monitor focused groundwater discharge to a sand-bed stream. *Water Resources Research, 52*, 4486–4503. doi: 10.1002/2016WR018808

Sharp, Z. (2007). *Principles of Stable Isotope Geochemistry*. Upper Saddle River, NJ: Pearson Prentice Hall.

Tockner, K., & Stanford, J. A. (2002). Riverine flood plains: Present state and future trends. *Environmental Conservation, 29*(3), 308-330. doi: 10.1017/S037689290200022X

Winter, T. C., Harvey, J. W., Franke, O. L., & Alley, W. M. (1998). *Ground Water and Surface Water: A Single Resource* (US Geological Survey Circular 1139).

---

## Author Comment (AC2) · 18 Mar 2019

**Reply to Reviewer #2**

*1. While the paper points out challenges in measuring gw-sw interaction, including a nice summary table by method, the challenges do not link up with the specific issues in braided rivers. In other words, the challenges would apply to all river types. For instance, the authors mention how heterogeneity makes it difficult to measure flow. While braided streams may be more heterogeneous (however that is defined), all streams would benefit from methods that address heterogeneity. I would have liked to see how the cross-sectional heterogeneity (in contrast to along reach) impacts measurement techniques. That said, Genereux's group has some papers illustrating both along-reach and cross section variation in streambed K, so even this aspect is not unique to braided rivers. The advantages and disadvantages sections list challenges that would apply to other stream types as well. The abstract and conclusions emphasize the need for multiple methods and consideration of scale in selecting methods, but again these recommendations apply to any river type. Without details about why a particular method works elsewhere but not on braided rivers, the paper lacks focus. It does not suffice to say a method is "more difficult" when it is difficult in a variety of river settings.*

Response: Thank you for the constructive feedback. We agree that many of the challenges of conducting studies in braided rivers are also present in other river environments. However, in braided rivers these challenges tend to occur to a larger degree. We have attempted to convey this within the introduction and particularly in the paragraph at L60. However, we agree that this point can be made with greater clarity and we intend to revise the introduction in order to do so.

During the process of writing the review paper, we critically examined all published studies we could find on measuring groundwater-surface water interactions in braided rivers. We looked at all of the techniques used in these studies and attempted to summarise for the reader what was effective and what was not. In terms of a particular method that works elsewhere but likely not in a gravel-bed braided river, we specifically mention seepage meters in L773-778. However, we believe we can improve the manuscript in this respect, for example we can discuss how there have been various designs of mini-piezometers used in rivers, but many of these designs would be unlikely to be effective in gravel-bed braided rivers, which is why we point the reader to studies where they have been deployed successfully. Flow gauging for calculating catchment or reach-scale water budgets is another method that may be effective in

other river environments but is very challenging in braided rivers. For example, because of the errors associated with flow gauging it is very difficult to estimate losses or gains in reaches with a sufficient degree of accuracy. Flow gauging does not differentiate between groundwater inflow and hyporheic water re-emerging in streams, and given the significant portion of flow in braided rivers that occurs within the riverbed, this is a considerable issue. As another example, it is very difficult to take undisturbed core samples for hydraulic conductivity tests when the river substrate contains coarse gravels, as gravel-bed braided rivers do.

*2. The paper provides a map of locations with braided streams, but does not justify why these locations are included and not others. The definition of what "concentrated" means in terms of distribution of braided streams is not provided. There is a list of braided streams in the US on [https://commons.wikimedia.org/wiki/Category:Braided_rivers_in_the_United_States_by_state](https://commons.wikimedia.org/wiki/Category:Braided_rivers_in_the_United_States_by_state), which suggested that braided rivers are important in the US too, yet no sites there are listed. To list the map as a significant feature of the paper ("to the authors' knowledge, this is the first map of its kind") but provide no details on how the map was generated is frustrating to the reader.*

Response: Thank you for your feedback on Fig. 1. We agree that the term "concentrated" is ambiguous and we will replace it in the text at L27 and L881 with "mainly found". This conforms to wording used in our justification for only displaying a selection of regions where braided rivers occur, at L64-66: "There are instances of braided rivers at locations outside of these regions (e.g., Russia, U.S., Scotland), however these locations are not shown in Figure 1 because, at a global scale, they are not where braided rivers are mainly found." To further justify our selection of locations displayed, we will add the following sentence to the manuscript at L64: "The regions displayed in Figure 1 are regularly cited in literature on braided rivers as the main regions where this river type can be found (e.g., Tockner et al., 2006; Hibbert & Brown, 2001)." We propose to add Russia to Fig. 1 based on comments in studies (Chalov & Alexeevsky (2015), Alexeevsky et al. (2013)) about the high number of braided rivers in the country.

In regard to braided rivers in the United States, we respectfully point to L64, where we did mention the U.S. containing braided rivers. Thank you for pointing to the Wikipedia link (https://commons.wikimedia.org/wiki/Category:Braided_rivers_in_the_United_States_by_state) , however it does not appear that this is a reliable authority on instances of braided rivers in the U.S. The webpage is a list of user-generated images that have included tags with the wording "braided river", however many of the images shown here do not appear to be of braided rivers (e.g. on the pages for Alabama, Virginia, Utah and Massachusetts).

*3. The word "hyporheic" only appears in the abstract and end of the paper, not in the main body. This mention in the abstract should be removed since it is not a topic covered in the paper. It is probably better left to another paper as the issues in measuring hyporheic flow differ significantly.*

Response: Thank you for your comment, however the term "hyporheic" was mentioned 16 times throughout the paper, and it was specifically discussed in conjunction with several of the cited studies. We believe this review would be significantly lacking if we did not include information on hyporheic flow processes. As we have noted at L100, issues surrounding hyporheic zone processes are of great importance in the management of braided rivers. There is a significant amount of river flow that occurs within the hyporheic zone in braided rivers. Further, as highlighted at L377, it is often difficult to distinguish between regional groundwater discharge into rivers and re-emerging river water from the hyporheic zone. While this is an issue that would be faced in other river environments, it is likely that this is more of an issue in braided rivers which 1) have highly permeable river bed strata, and 2) have a significant amount of river flow that occurs within the streambed. As Referee #1 highlighted (see comment #9), we need to clarify what we mean by hyporheic flow and the hyporheic zone, as well as better explain the related research gaps. Please refer to our response to Referee #1 (comment #9) for more details on how we intend to clarify this in the revised manuscript.

*4. The modeling discussion is focused too much on MODFLOW. The description of MODFLOW packages can be found elsewhere and there are other models that incorporate groundwater-surface water interaction that could be discussed. For example, a recent special issue in Groundwater on integrated modeling included a paper on streambed heterogeneity. There is also a recent review paper on modeling gw-sw interaction in Reviews of Geophysics*

*that provides a broader view. The abstract mentions the need for new approaches in modeling, but the paper does not provide sufficient direction to justify this as a conclusion of the paper. The conclusion the models need more data and more sensitivity analysis has been stated many times before.*

Response: In the review we focussed on MODFLOW as this is the code that most previous studies have used to model groundwater-surface water exchange in braided rivers. We agree that the description of MODFLOW packages can be found elsewhere and in the revised manuscript we will remove details of these. Sentences from L676 to 687 will be replaced with the following: "Several packages are available in MODFLOW for simulating surface water-groundwater interaction and further details about the application and limitations of these can be found in Brunner et al. (2009, 2010).

We agree that it would be helpful to include specific recommendations on new approaches to modelling including codes (such as HydroGeoSphere) and methods such as those detailed in Brunner et al. (2017).

*5. I was surprised that fiber optic temperature systems (also known as DTS for distributed temperature systems) and geophysics were not discussed. These methods have been mentioned in other reviews and provide broader coverage which might benefit braided streams. I found it odd to bring up thermal imaging for the first time in the discussion section rather than in the review of methods, especially since it is mentioned in the abstract and it is one of the more promising techniques for heterogeneous systems. An example of the benefits of thermal imaging might provide an interesting figure.*

Response: Thank you for your suggestions. We agree that the manuscript would benefit from discussing geophysical techniques and DTS as possible methods to apply in braided rivers. We have replied to this in more detail in our response to Reviewer #1 (see Comments #4, 7, 23 and 50). We will add more detail on additional methods that can be applied in an expanded "Key gaps and possibilities" section of the revised manuscript.

*6. On the topic of figures, the figures were lacking in illustrative examples of applications. There was a map, but the other figures were photos or diagrams and didn't show quantitative*

*challenges or opportunities. In other words, I think it would help the readers' understanding to include data figures.*

Response: In the revised manuscript we will consider adding additional figures that may be helpful to illustrate the techniques discussed.

*7. One place that the paper focuses on braided streams is the literature review of methods. The paper summarizes applications in braided streams and the table of methods lists braided stream citations. However, the literature summary sections of the paper are a bit dry. They list highlights of each paper one after another. I think some of these papers could be describing non-braided streams and the reader would not know. This type of literature review needs to be briefer and provide synthesis of issues specific to the problem identified. In addition, a significant number of references (estimated 25% based on the first page of the bibliography) are not readily available literature but reports or theses (typically from NZ). Many readers will not have ready access and the focus on one region is not justified.*

Response: We agree that it would be beneficial to shorten the sections that discuss prior literature, and we intend to do this in the revised manuscript.

We endeavoured to make it clear through the introduction that the review focusses on braided rivers, so that if there were other types of rivers discussed, we would specifically note this. We will add a sentence to this effect in the introduction, making this explicit.

In regard to technical reports and theses that have been cited, we apologise for not including URLs in the reference list and will fix this in the revised manuscript. We will also remove the following references that are not publicly available: Anderson (2004), Davey (2004), Aichison-Earl & Ritson (2013), Williams & Aitchison-Earl (2006).

We realise that there is a heavy weighting on studies conducted in New Zealand and this is not intentional by any means. The vast majority of published studies on this topic that we were able to find were based in New Zealand. In fact we chose not to discuss several New Zealand studies in the methods section as we felt studies from this country were over-represented. Through the process of gathering literature, we did consider possible reasons for the apparent over-representation of New Zealand studies. We considered that search engine results may have been weighted to display New Zealand studies as the authors were based in

New Zealand. To assess whether this was the case, in addition to more general searches, we specifically searched for literature in countries or regions (e.g., Italy) where braided rivers are common. This search method produced some, but not many, additional relevant references. Further, the majority of search engines used were via scientific indexing sites (e.g., Web of Science), which to the authors' knowledge, do not tailor search results in the way that Google does. We do acknowledge that we are likely to have missed literature published in languages other than English, but this issue is likely not unique to our review.

*8. It can be difficult to meet the standards of a review article. In the end, I ask myself whether I would give this paper to colleagues to read, or just keep recommending Kalbus et al. or LaBaugh and Rosenberry as review papers on the topic. I do not think there is enough new material here for me to consider this paper to be an update on the earlier papers. If revising, I would recommend a very short review paper, which introduces Table 1 and gives the reader the reference list for readers to select topics on their own (rather than the one line summaries of each paper). The shorter paper also needs to provide the reader with an approach to braided streams that is distinctly different than other streams – this message will take additional synthesis and thus I would consider it to be a new paper rather than a resubmission. Hence, I am recommending rejection and significant redirection for any new submittal.*

Response: We are grateful to the reviewer for their constructive comments. While there have been a number of review papers on surface water-groundwater interaction, none have focused on braided rivers previously and this is the gap we are wanting to address. We would argue that braided rivers have features that are unique enough to warrant a review paper that focuses on this river type specifically. As detailed above, we intend to revise the introduction to clarify the unique characteristics of braided rivers. We will shorten the review of previous studies in Section 3 and increase our use of Table 1 to provide guidance to the reader, as suggested. Also, we will work on expanded the "Key gaps and possibilities" section to enhance the novelty of the paper by suggesting emerging and promising techniques being used in other environments and that are likely to have application in braided rivers.

We have had very positive feedback on this manuscript from numerous groundwater researchers and managers within our networks and we look forward to using the guidance

provided by the reviewer to improve the manuscript. We are confident that we can achieve that as part of the current submission.

**References**

Aitchison-Earl, P., & Ritson, J. (2013). Surveys of groundwater level and river flow in 2010-2011 from the Rakaia River to the Ashburton River/Hakatere. Christchurch, New Zealand: Environment Canterbury. Report No. R13/26.

Alexeevsky, N. I., Chalov, R. S., Berkovich, K. M., & Chalov, S. R. (2013). Channel changes in largest Russian rivers: Natural and anthropogenic effects. *International Journal of River Basin Management, 11*(2), 175-191. doi: 10.1080/15715124.2013.814660

Andersen, B. (1994). Groundwater between the Selwyn and Rakaia rivers. Masters Thesis. University of Otago, Dunedin, New Zealand.

Brunner, P., Cook, P. G., & Simmons, C. T. (2009). Hydrogeologic controls on disconnection between surface water and groundwater. *Water Resources Management, 45*(1). doi: 10.1029/2008WR006953

Brunner, P., Simmons, C. T., Cook, P. G., & Therrien, R. (2010). Modeling Surface Water-Groundwater Interaction with MODFLOW: Some Considerations. *Ground Water, 48*(2), 174-180. doi: 10.1111/j.1745-6584.2009.00644.x

Brunner, P., Therrien, R., Renard, P., Simmons, C. T., & Franssen, H.-J. H. (2017). Advances in understanding river-groundwater interactions. *Reviews of Geophysics, 55*, 818–854. doi: 10.1002/2017RG000556

Chalov, S. R., & Alexeevsky, N. I. (2015). Braided rivers: Structure, types and hydrological effects. *Hydrology Research, 46*(2), 258-275. doi: 10.2166/nh.2013.023

Davey, G. (2004). Stream Depletion in the Ohapi Creek Catchment. Christchurch, New Zealand: Environment Canterbury. Report U04/55.

Hibbert, B., & Brown, K. (2001). *Braided River Field Guide.* Twizel, New Zealand: Department of Conservation and Meridian Energy Limited. Retrieved from https://www.doc.govt.nz/globalassets/documents/conservation/land-and-freshwater/freshwater/prr/braided-river-field-guide.pdf

Tockner, K., Paetzold, A., Karaus, U., Claret, C., & Zettel, J. (2006). Ecology of braided rivers. In G. H. S. Smith, J. L. Best, C. S. Bristow & G. E. Petts (Eds.), *Braided Rivers:*

*Process, Deposits, Ecology and Management* (Vol. Special Publication Number 36 of the International Association of Sedimentologists). Malden, MA, USA; Oxford, UK; Carlton, Victoria, Australia: Blackwell Publishing.

Williams, H., & Aitchison-Earl, P. (2006). Relationships between groundwater pressures and lowland stream flows in the Lake Ellesmere area. Christchurch, New Zealand: Environment Canterbury. Report Number U06/31.

---

## Author Response (AR1)

[revised manuscript text omitted]

2002; Chen 2007; Passadore et al., 2015; Scott and Thorley, 2009; Baalousha, 2012; Wilson and Wöhling,

2015; Wöhling et al., 2018)..

. Shu and Chen (2002) developed a transient MODFLOW model to understand the spatial and seasonal variation in surface water-groundwater exchange for the braided Platte River, in Nebraska, USA, one of the most well- known braided rivers in the United States. Shu and Chen (2002) also simulated pumping from over 1000 wells to better understand the relationship between seasonal pumping from irrigation wells and river levels.

Simulation results suggested that continued over-extraction of groundwater in the region would gradually increase losses from the river. In a subsequent study, Chen (2007) developed a numeric model using the

Galerkin finite-element method to assess the impact of the riparian vegetation on baseflow interception and increased river water infiltration in the Platte River.

Passadore et al. (2015) developed a three-dimensional groundwater flow model for the large and complex

Central Veneto aquifer in northeast Italy, which is highly connected to the braided Brenta River. The authors calibrated the model using historic observation data from groundwater wells. They first created a conceptual geologic model, which was used to set up a finite-element numeric model. They used aquifer parameter inputs (e.g., hydraulic conductivity, transmissivity) based on field observations and carried out a steady-state calibration based on observations from 100 wells. The authors provide no detail regarding the manner in which the Brenta River was implemented in the model but note that fluxes from the river to the underlying aquifer are an important component of aquifer recharge.

Ramanathan et al. (2010) examined the heterogeneity in braided river deposits using the geometric-based simulation method, and the resulting model can be used to simulate fluid flow. Their model incorporated typical braided river features such as areas of lower permeability, high permeability structures such as open-framework gravels, and channel shifting dynamics.

There are several braided river studies conducted in New Zealand that involved modelling groundwater-surface water exchange. In the Canterbury region, Anderson (1994) and Scott and Thorley (2009) developed a relatively simple steady-state MODFLOW model,s for the purpose of improving understanding of regional-scale water budgets, which included gains and losses from the braided Rakaia, Selwyn and Waimakariri Rivers. In Hawkes

[revised manuscript text omitted]

 **Reply to Reviewer # 1**

 We thank Reviewer #1 for the helpful review and have responded to each comment below.

 All line numbers in our responses refer to the revised marked manuscript.

 **General Comments**

 *1.     The manuscript "A review of methods for measuring groundwater-surface water*

 *exchange in braided rivers" by Katie Coluccio and Leanne Kaye Morgan is a review paper.*

 *As the title suggests it is about measuring methods for groundwater-surface water exchange*

 *in braided rivers. In general, the manuscript is informative, provides an overview about the*

 *current literature, is well structured and well written.*

 Response: We thank the reviewer for these positive comments.

 *2.     However, some sections are lengthy and might be shortened.*

 Response: We have shortened the descriptions of the literature as suggested in Comment #6,

 which has resulted in approximately 2,800 words being deleted. However, text has been

 added as a result of addressing the various comments by both reviewers, particularly in

 respect to discussing additional methods that could be used in braided rivers to investigate

 groundwater-surface water exchange.

 *3.     Furthermore, as indicated in the major comments below important information,*

 *definitions etc. is missing.*

 Response: We thank the review for pointing out these omissions. As detailed below (in

 response to comments #8, 9, 11, 12, 13, 16, 17, 18, 19, 22, 23, 24, 26, and 27), we have

 addressed this in the revised manuscript.

 *4.     In general the authors could think a little bit more outside of the box. They are very*

 *focused on the methods that have been used in studies of groundwater-surface water*

 *interactions in braided rivers. But there are several similar groundwater-surface water*

 *interfaces and as part of a scientific paper I would expect the authors to consider additional*

*methods that might be adapted to braided rivers in future in addition to simply summarising*

*the literature available at present.*

Response: In the revised manuscript we have discussed (in Section 4 "Key gaps and possibilities") additional methods that have not yet been applied in braided rivers but show potential. Here we have included additional temperature methods such as fibre-optic temperature sensing, active heat pulse methods and thermal infrared imaging; artificial tracers (such as dye, salt and bacterial tracers); remote collection of data via satellite imaging and unmanned aerial vehicles; geophysical techniques; and modelling packages (i.e.,

HydroGeoSphere and MIKE-SHE).

*5.     I think the manuscript can be published after revision.*

Response: Thank you.

**Major comments**

*6.     Entire manuscript: Try to shorten your manuscript and avoid lengthy descriptions of*

*the literature, e.g. L173-L213, L216-242, L289-325, L328-L379, L382-402, L533-603,*

*L606-L640, L667-L739.*

Response: We agree that the manuscript would benefit from more concise descriptions of the literature and have shortened sections at L204-214, 220-229, 256-276, 280-284, 336-352,

360-365, 384-387, 392-407, 412-417, 445-452, 457-459, 467-476, 512-543, 548-564, 585-

589, 641-643, 655-656, 661-663, 694-695, 700-704, 749-760, 768-803, and 808-816. This has resulted in the deleting approximately 2,800 words from the original manuscript.

*7.     L60 & entire manuscript: Suggest also methods that have been successfully used at*

*other groundwater-surface water interfaces and that might be adapted to braided rivers and*

*might be used in braided rivers in future. Reporting only what has already been done in*

*braided rivers is a little bit thin.*

Response: Thank you for the suggestion. Please refer to our response to Comment #4.

8.      *L64 & Fig. 1 & L882: I strongly recommend adding all additional instances of*

*braided rivers outside of the major regions. You might use different symbols for major*

*regions with braided rivers and single instances.*

Response: When initially creating Fig. 1, we had considered attempting to include all instances of braided rivers globally, as suggested. However, we decided against this for a few reasons. Mainly, we were concerned that stating we had accounted for "all" braided rivers would run the risk of missing some rivers and in so doing being factually incorrect. Secondly, we felt that highlighting the locations where most braided rivers occur would be most useful to readers, as this indicates where most of the braided rivers research has been conducted. In an attempt to account for instances of braided rivers outside of the major regions, in the revised manuscript, we have added a sentence at L72 noting that braided rivers also occur in small numbers in the U.S., Scotland, Iceland, China, Poland, Belarus, Colombia, Congo,

Brazil, Paraguay, Argentina, and the Touat Valley in Africa. Also, we have added Russia to

Fig. 1 based on comments in studies by Chalov & Alexeevsky (2015) and Alexeevsky et al.

(2013) about the high number of braided rivers in that country.

9.      *L100f; L791f, L855: I think there is a need for clear definitions of "groundwater-*

*surface water interactions" and of "hyporheic exchange". Often, the term "groundwater-*

*surface water interaction" is used in literature in a wide sense including hyporheic exchange*

*as one process of groundwater-surface water interactions. However, according to line 100f*

*you consider both as separate processes with some impacts on each other.*

Response: Thank you for highlighting this, and we agree that this is an area where more clarity would be helpful. We have added the following text to the revised manuscript at L60: "This paper often refers to groundwater-surface water exchange, which in this context may include regional groundwater exchange with river water, as well as hyporheic zone exchange. Researchers have defined the hyporheic zone and the exchange processes that occur there in many ways (e.g., Krause et al., 2011; Cardenas, 2015). In the present paper, hyporheic exchange refers to downwelling or upwelling of water through the hyporheic zone, i.e., the saturated area between the streambed and shallow aquifer where stream water and shallow groundwater mix.

10.     *L134ff: Even though I agree that there is little research about groundwater-surface*

*water interactions in braided rivers your "Web of Science" search is meaningless. I tried to*

*reproduce it. First of all "groundwater and surface water interactions" with "…" results in*

*much smaller numbers than the ones reported by you, e.g. only three papers for lakes instead*

*of 437 reported by you. Repeating the search without "…" resulted in approximately the*

*numbers reported by you. However, having a closer look at those papers revealed that most*

*of the hits are not about groundwater-surface water interactions at all but that the separate*

*words of the phrase are used in separate sentences and in different context. Furthermore, at*

*many of the interfaces mentioned by you (lakes, ocean, stream) specific terms are used, e.g.*

*"lacustrine groundwater discharge", "submarine groundwater discharge" and "hyporheic*

*zone" instead of "groundwater and surface water interactions". Sometimes the word*

*"interactions" is substituted by "exchange" or by "interfaces". Also, there are different*

*spellings for "groundwater" such as "ground water". I am quite sure that the largest*

*number of studies focusing on groundwater-surface water interactions is about stream,*

*followed by (coastal) oceans followed by lakes and finally by braided rivers. You might also*

*have a look at review papers focusing on the different interfaces. There are several of them. I*

*recommend either deleting lines 134-139 or repeating this literature search with a set of*

*different keywords to get a more comprehensive overview of the literature of interest.*

Response: Thank you for highlighting this issue, and we agree that deleting these lines would improve the manuscript. They have been removed from the revised manuscript (L150–155).

*11.      L158ff: From my experience budgets are often quite error-prone because accurate*

*measurements of river discharge are challenging. Often changes in river discharge between*

*stations are much smaller than the error inherent to the measurements. You should mention*

*this shortcoming more clearly than only in lines 261-263.*

Response: Indeed, this is an important factor to consider. Additional to L297, we have mentioned this limitation in Table 1 (under Water Budgets), so we believe this limitation has been adequately addressed.

*12.      L272ff/L284ff: I think it is important to introduce here also the concept that tracers*

*need to be conservative (on the scale of the investigation). In this context, I doubt that*

*dissolved oxygen (L284), nitrate (L285), sulphate (L286) and pH (L404) are useful tracers.*

*pH might be acceptable in the context of alkanity but that also needs more discussion. The*

*concentrations of oxygen, nitrate, sulfate and H+ will be altered due to many different*

*biogeochemical processes. They might be used under certain circumstances and on small*

*scales on which little turnover takes place. But this is something very critical. If you list these*
*compounds you need to discuss them critically.*

Response: Thank you for the comments here and we agree with your point that tracers need
to be conservative, and this is an important consideration to make when selecting parameters
to measure. In the revised manuscript, we have added comments to this effect in the
Advantages & Limitations section of Section 2.2 as well as in the Hydrochemistry section of
Table 1. We think that these parameters are still worthy of discussion as they have been used
in several previous studies to varying degrees of success.

*13.     L272ff: In addition to environmental tracers I recommend to discuss also artificial*
*tracers that might be added to the system. There are multiple studies using artificial tracers*
*and I am quite sure that the also have been used in braided rivers. However, even if not they*
*are an option that should be considered.*

Response: In Section 4 of the revised manuscript, we have added a paragraph on the use of
artificial tracers such as dyes, salt and bacteria. Here, we have cited several studies conducted
in non-braided river environments: Binley et al., 2013; Ferreira et al., 2018; Stoner et al.,
2013; Knöll and Scheytt, 2018; González-Pinzón et al., 2015. We have also cited several
studies that used artificial tracers to characterise alluvial aquifer properties in a well array in
New Zealand: Close et al., 2002; Dann et al., 2008; Sarris et al., 2018.

*14.     L457-468: I don't see any connection of this paragraph to the topic groundwater-*
*surface water interactions. Therefore, I recommend deleting this paragraph.*

Response: Thank you for highlighting this, and we agree that this study was not specifically
related to investigating groundwater-surface water interactions, and thus we have removed it
from the revised manuscript (at L516-528).

*15.     L469-484: The topic of the present review is measurement methods for groundwater-*
*surface water interactions. Thus, these two paragraphs don't fit to the topic of the review*
*paper. They are about impacts of groundwater and surface water on temperature (and*
*ecological consequences) but not how to use measurements to identify groundwater-surface*
*water interactions.*

Response: Thank you for the constructive comments on these studies. We have deleted these
two paragraphs (L529-543). The two studies mentioned in these paragraphs (i.e., Acuna and
Tockner, 2009; Malard et al., 2001) used multiple methods to assess groundwater-surface
water exchange, and thus we feel that these are useful references to include, albeit now only
in Table 1 within the revised manuscript.

*16.    L502ff: I think it is important to measure temperature depth profiles as you do in this*
*paragraph. However, you should go into a little bit more detail here and also mention typical*
*evaluation methods for temperature depth profiles such as the steady state approach (e.g. C.*
*Schmidt, M. Bayer-Raich, and M. Schirmer. Characterization of spatial heterogeneity of*
*groundwater-stream water interactions using multiple depth streambed temperature*
*measurements at the reach scale. Hydrology and Earth System Sciences 10:849-859, 2006)*
*or VFLUX.*
Response: We agree that it would be useful to include some more detail on how temperature
depth profiles may be analysed. VFLUX was used in the Coluccio (2018) study, and a note in
this regard has been added to the revised manuscript at L561. Both VFLUX and the steady
state method used by Schmidt et al. (2006) have also been mentioned at L507-508 in the
revised manuscript.

*17.    L443ff: I think at one point in this subchapter you should clearly differentiate between*
*methods that are used to determine fluxes (e.g. temperature depth profiles) and methods for*
*pattern identification (aerial TIR, fo-DTS). This applies also to lines 513-515. TIR is a*
*method for pattern identification. However, you need to describe this already before and not*
*only in Advantages and Limitations. See also comment regarding this topic below.*
Response: This is a very good suggestion, thank you. In the revised manuscript we have
noted this difference in temperature methods at the beginning of section 2.3 (L508-512).

*18.    L443ff: Furthermore, you should briefly mention typical approaches to measure*
*temperature and in this paragraph you should also include fibre-optic distributed*
*temperature sensing even if it has not been used in braided rivers yet.*

Response: We agree that it would be useful to have a brief explanation of typical approaches to measuring temperature while noting which ones are used for pattern recognition or flux estimates (as per the comment above). We have briefly mentioned fibre-optic DTS in the beginning of section 2.3 (L509) and discussed it further in section 4 (L982-983), as well as included relevant references.

19. *L443ff: You could also consider adding temperature methods that don't rely on natural temperature differences but use temperature as an active tracer, e.g. active (heated) DTS, heat-pulse sensors etc.*

Response: Thank you for the suggestion, and we agree that it would be useful to include active heat tracers (such as the 3D heat pulse array used in Banks et al. (2018)) in the revised manuscript. To our knowledge, these methods have not yet been used in a braided river setting, but they do have potential and thus may be beneficial for readers. We have included these methods in section 4 of the revised manuscript (L984).

20. *L524: "Hydraulic property measurements" is no suitable chapter headline for the subchapter "Groundwater observation wells"! Alternatives might be "2.4 Flow-net analysis" or "2.4 Darcy approach". I would call 2.4.1 "Hydraulic gradients" and 2.4.2 "Hydraulic conductivity".*

Response: Thank you for the suggestions and we have amended the section headings to the following:

2.4 Darcy approach

2.4.1 Hydraulic gradient

2.4.2 Hydraulic conductivity

Advantages and Limitations

21. *L525ff: The second sentence of the paragraph is wrong: The groundwater level/hydraulic gradient is no hydraulic property. Hydraulic properties are the hydraulic conductivity, the porosity etc. The rest of the paragraph belongs to 2.4.2.*

Response: Thank you for highlighting this. We have deleted L585-589 as these points are
covered in sections 2.4.1 and 2.4.2. This has also served to shorten the manuscript.

*22.    L559ff: You use the terms well, piezometer and mini-piezometer but I have not seen a*
*definition of those terms. Consider to include also other designs, e.g. M. O. Rivett, R. Ellis, R.*
*B. Greswell, R. S. Ward, R. S. Roche, M. G. Cleverly, C. Walker, D. Conran, P. J. Fitzgerald,*
*T. Willcox, and J. Dowle. Cost-effective mini drive-point piezometers and multilevel samplers*
*for monitoring the hyporheic zone. Quarterly Journal of Engineering Geology and*
*Hydrogeology 41:49-60, 2008. However, in this paragraph with its focus on groundwater*
*level measurements either sufficient diameter for a logger or an electric contact gauge is*
*useful even though some scientists used innovative approaches for very small diameters*
*(transparent tubes, suction to increase water level differences to an easily visible height,*
*colored strings ...) Also, you should consider describing at least in brief typical installation*
*techniques for the different designs and different depth depending on substrate quality.*
*Furthermore, report at least in one sentence how water tables are measured/logged.*
Response: In L619-623 we intended "groundwater well" and "piezometer" to be
synonymous. To clarify this, we have modified the sentence to read: "In terms of specific
methods that can be used for measurements, existing piezometers (i.e., monitoring wells) near
rivers can be useful for conducting these types of studies, particularly given the high cost of
drilling new wells." At L623 we have added the sentence: "Please refer to standard text such
as Fetter (2001) for a definition of piezometers". In L623-625 we have defined "mini-
piezometers" as "scaled-down versions of piezometers and typically installed no deeper than
about two metres". With respect we prefer not to include reference to installation methods as
these are detailed in the cited references. Also, for the sake of brevity, we prefer not to detail
other piezometer designs.

In the revised manuscript, we have commented in the Advantages and Limitations section of
2.4 about the need to consider the diameter of wells being used with downhole equipment
such as loggers. Also, at the beginning of section 2.4.1 we have briefly detailed the way in
which water levels are typically measured.

*23.    L605ff: Consider to add also in brief the use of geophysics to characterize the*
*subsurface pattern (together with some core for calibration of geophysical methods).*

Response: This is a good suggestion, thank you. In the revised manuscript, we have discussed new methods for use in braided rivers in section 4. For the sake of brevity, we discuss geophysics in that section (L1002-1014).

*24.    L642ff: Mention that loggers require a certain diameter of wells/piezometers as a*
*further disadvantage.*

Response: At L716-717 we now include the following: "Another important factor to consider is that many data loggers require a certain diameter well."

*25.    Table 1: You have split the first method (water budget) into two budget methods. Why*
*haven't you also split the following methods as in the text (e.g. environmental tracers, heat*
*tracers, …). In fact heat tracers are also an environmental tracer. Why are River reach*
*budgets suitable only for relatively homogenous aquifers? Remove pH and DO from*
*environmental tracers (see corresponding comments above). As far as I understand the table*
*and its table capitations it is about methods for quantifying water fluxes. The point "Aerial*
*surveys can be faster than in-stream surveys" does not fit. This is a method for pattern*
*identification and not for flux determination. As described above I doubt that "Hydraulic*
*Property Measurement" is an adequate headline for this type of method. I don't think that*
*this applies only to minipiezometers. Piezometers are also easy and quick to install. In*
*general other authors have grouped their methods into three categories and I think this*
*would be advantageous here as well:*
*+ point methods to estimate fluxes at a discrete location*
*+ methods for pattern identification don't yield numbers for fluxes but can help to identify*
*representative sites and the most extreme sites to conduct the point methods at the most*
*interesting sites. Under certain circumstances also transfer functions possible that combine*
*methods for pattern identification and point methods*
*+ integrating methods over large areas that result in total fluxes, but without any information*
*about local fluxes or distribution of patterns.*
Response: Thank you for your thorough comments on Table 1. The intention of this table was to summarise all the methods discussed in the review, both for identifying patterns and for estimating fluxes. Perhaps the table title has created the confusion here, so we have amended the title to read "Advantages and disadvantages of various methodologies for **measuring** groundwater-surface water interactions in braided rivers".

We are not convinced that organising the methods according to scale of measurement would be helpful as there would be overlap amongst methods (i.e., some methods could be used at multiple scales, see Fig. 1 in Kalbus et al. (2006)). We have revised the categories for Table 1 to: Water budgets, Hydrochemistry, Temperature studies, Darcy approach and Modelling.

*26.     L783ff: Please keep the three points above in mind. Remote sensing is not gathering the same information as the point methods mentioned in L781-783! The same applies to Line 870-872.*

Response: Thank you for highlighting that we may need more clarity around scales of measurement. However, we are not sure why there is confusion here. Depending on how they are carried out, the methods mentioned in L870-871 (pumping tests, flow gauging, stable isotope analysis and hydrochemical tracers) can provide broad spatial scale information, as can TIR imaging, geophysical methods and satellite data. As we mentioned in the response to comment #25 above, many of these methods can be used to collect data at various scales, while some indeed are point methods only (e.g. permeameter tests or 1-D temperature profiles).

*27.     L797f: Please mention here also that time series that might be recorded with loggers can be very useful to gain system understanding because groundwater-surface water interactions might vary with time and even the flow direction might reverse over time.*

Response: Indeed, this is an important point to make. In this section of the manuscript, we intended to illustrate this with the example of temperature time series data in L890-891, but in the revised manuscript we have added an additional sentence here to make it clearer that time series data for a range of parameters can be very useful to observe changes in groundwater-surface water interactions over time. The added sentence at L892 reads: "In addition to temperature, many parameters can be collected as time series (e.g., water levels, hydrochemistry), which may be very useful for interpreting temporal changes in groundwater-surface water exchange."

28. *L849: It is definitely strange to have a subchapter 3.1 but no 3.2. Also, it is confusing that the introduction before 3.1 is about 5 pages long and 3.1 less than 1 page long.*

Response: Thank you for highlighting this. We have changed the numbering of Section 3.1 (Key gaps and possibilities) to Section 4.

**MINOR COMMENTS**

29. *L48: Cite also Winter et al. (1998) (https://pubs.usgs.gov/circ/circ1139/)*

Response: Thank you for the relevant suggestion. This reference has been added to the revised manuscript (L48).

30. *L102f: Why is improved knowledge of historical patterns needed? In addition, can you please cite a reference.*

Response: Better knowledge of historic states and patterns of braided rivers would be very helpful for understanding the implications of modifications to natural systems in order to set water allocation limits and minimum flow levels in rivers (Riegler, 2012; Burbery et al., 2010). For example, many irrigation schemes have artificially raised groundwater levels due to land surface recharge, or lowered groundwater levels due to abstraction in comparison to their natural (pre-irrigation) states. In some rivers this has affected the losing/gaining patterns. A comment in this regard (with references) has been added to the revised manuscript.

31. *L118: A more scientific reference would be great here.*

Response: The reference has been replaced at L134 by references to Caruso (2006); Larned et al. (2008); Tockner and Stanford (2002), which are all peer-reviewed publications in international journals.

32. *L147: Consider adding Rosenberry et al. (2015) (https://onlinelibrary.wiley.com/doi/full/10.1002/hyp.10403)*

Response: This is a very useful reference, and we have added it to the revised manuscript,
along with Brunner et al. 2017, which is a useful review of the latest advances in methods for
characterising and modelling river and groundwater interactions and specifically mentions
braided streams in some parts.

*33.    L279: I think what is much more important than evenly distributed groundwater*
*discharge or recharge is an even groundwater concentration.*
Response: Thank you for the suggestion, and we agree that amending the wording would be
more accurate. The wording at L313-315 has been changed to "This type of analysis assumes
there is an evenly distributed groundwater concentration between sampling locations and that
there is complete mixing of water sources."

*34.    L289ff: Please correct: there are three stable oxygen isotopes including O-17!*
Response: Thank you for highlighting this oversight. Oxygen-17 has been added to this
discussion of stable oxygen isotopes at L326.

*35.    L291f: "The process is largely driven by temperature, whereby … at higher elevation*
*due to colder temperatures" The process is not driven by elevation but the elevation effect is*
*a result of decreasing temperatures with increasing depth. In case you really want to mention*
*processes in addition to temperature you can add humidity and salinity as further processes.*
Response: Thanks for pointing this out. At L328, we have amended this sentence to read:
"The process is largely driven by temperature, humidity and salinity, whereby precipitation is
increasingly depleted in $^{18}$O at colder temperatures (which tend to occur at higher elevations)
(Sharp, 2007)".

*36.    L519: I think the most important point that should be measured here is season!*
Response: Thank you, this has been added to the revised manuscript at L580.

*37.    L560: Only deep wells/piezometers are expensive.*

Response: Agreed. We have amended the text to reflect that cost of installing wells or piezometers may only be prohibitively high in some situations.

*38.    L565: Isn't this also a conceptual diagram of a well?*

Response: As detailed in our response to comment #22 above, we have modified the text at L619 so that it is clear that we consider piezometers to be monitoring wells. We have also made it clear that mini-piezometers are small versions of piezometers at L624. In light of this we think that the labelling of this figure as a conceptual diagram of a mini-piezometer is no longer confusing.

*39.    L656: You might want to mention that it is nearly impossible to take undisturbed cores/rings for KSat analysis if the sediment contains coarse gravel as this is the case in most braided streams.*

Response: Thank you. We have added a sentence to this effect at L728: "In particular, taking undisturbed cores of sediments containing gravel, as most braided rivers do, is nearly impossible."

*40.    L 674: "interactions" instead of "interaction"*

Response: Thank you. The text has been amended at L747.

*41.    L687: Delete: "and will not be repeated here."*

Response: Thank you. This text has been deleted at L760.

*42.    L704: "They used" instead of "The used"*

Response: The text at L783 has now been removed.

*43.    L754: You are not investigating groundwater and surface water but their interactions:*

*"… for investigation of groundwater-surface water interactions, and there …"*

Response: Thank you for recognising this error. The text has been amended at L842 to:

"There are many factors to consider when selecting the appropriate method(s) for studying groundwater-surface water interactions."

*44.     L764: "a study" instead of "the study"*

Response: Thank you. This change has been made at L852.

*45.     L808: "by the study objective and the study object*

Response: The text at L899 has now been removed.

*46.     L820: Only during storms???*

Response: This is a fair point. Our intention in specifically mentioning storms was to highlight mass sediment movement during flood events, but indeed, sediment transport at other times may equally damage equipment. The mention of storms at L911 has been removed.

*47.     L851: "One of the most …" – I do not understand this sentence.*

Response: In the revised manuscript we have amended L942-944 to the following: "There is limited understanding of how hyporheic flow processes operate, and how they impact river flow levels and water quality in braided rivers."

*48. L854: Consider adding here S. Krause, D. M. Hannah, J. H. Fleckenstein, C. M. Heppell,*

*D. Kaeser, R. Pickup, G. Pinay, A. L. Robertson, and P. J. Wood. Interdisciplinary*

*perspectives on processes in the hyporheic zone. Ecohydrology 4 (4):481-499, 2011.*

Response: This is an excellent suggested reference and has been added at L62 and L945.

*49.     L869: the present paper*

Response: This suggested text has been added at L960.

*50.     L895: You might add here DTS and geophysics*

Response: We agree, and we have added fibre-optic DTS and geophysics at L1049.

*summary table by method, the challenges do not link up with the specific issues in braided*

*rivers. In other words, the challenges would apply to all river types. For instance, the authors*

*mention how heterogeneity makes it difficult to measure flow. While braided streams may be*

*more heterogeneous (however that is defined), all streams would benefit from methods that*

*address heterogeneity. I would have liked to see how the cross-sectional heterogeneity (in*

*contrast to along reach) impacts measurement techniques. That said, Genereux's group has*

*some papers illustrating both along-reach and cross section variation in streambed K, so*

*even this aspect is not unique to braided rivers. The advantages and disadvantages sections*

*list challenges that would apply to other stream types as well. The abstract and conclusions*

*emphasize the need for multiple methods and consideration of scale in selecting methods, but*

*again these recommendations apply to any river type. Without details about why a particular*

*method works elsewhere but not on braided rivers, the paper lacks focus. It does not suffice*

*to say a method is "more difficult" when it is difficult in a variety of river settings.*

Response: Thank you for the constructive feedback. We agree that many of the challenges of conducting studies in braided rivers are also present in other river environments. However, in braided rivers these challenges tend to occur to a larger degree. We have attempted to convey this within the introduction and particularly in the paragraph at L151. However, we agree that this point can be made with greater clarity and we have revised the introduction in order to do so.

During the process of writing the review paper, we critically examined all published studies we could find on measuring groundwater-surface water interactions in braided rivers. We looked at all of the techniques used in these studies and attempted to summarise for the reader what was effective and what was not. In terms of a particular method that works elsewhere but likely not in a gravel-bed braided river, we specifically mention seepage meters in L862-

867. However, we believe we can improve the manuscript in this respect, for example we can discuss how there have been various designs of mini-piezometers used in rivers, but many of these designs would be unlikely to be effective in gravel-bed braided rivers, which is why we point the reader to studies where they have been deployed successfully. Flow gauging for calculating catchment or reach-scale water budgets is another method that may be effective in other river environments but is very challenging in braided rivers. For example, because of the errors associated with flow gauging, it is very difficult to estimate losses or gains in reaches with a sufficient degree of accuracy. Flow gauging does not differentiate between groundwater inflow and hyporheic water re-emerging in streams, and given the significant portion of flow in braided rivers that occurs within the riverbed, this is an important issue to consider. As another example, it is very difficult to take undisturbed core samples for hydraulic conductivity tests when the river substrate contains coarse gravels, as gravel-bed braided rivers do.

*2. The paper provides a map of locations with braided streams, but does not justify why these*

*locations are included and not others. The definition of what "concentrated" means in terms*

*of distribution of braided streams is not provided. There is a list of braided streams in the US*

*on*

*https://commons.wikimedia.org/wiki/Category:Braided_rivers_in_the_United_States_by_st*

*ate, which suggested that braided rivers are important in the US too, yet no sites there are*

*listed. To list the map as a significant feature of the paper ("to the authors' knowledge, this is*

*the first map of its kind") but provide no details on how the map was generated is frustrating*

*to the reader.*

Response: Thank you for your feedback on Fig. 1. We agree that the term "concentrated" is ambiguous and we have replaced it in the text at L27, L70 and L1034 with "mainly found".

This conforms to wording used in our justification for only displaying a selection of regions where braided rivers occur, at L71-75: "There are instances of braided rivers at locations outside of these regions (e.g., the U.S., Scotland, Iceland, China, Poland, Belarus, Colombia,

Congo, Brazil, Paraguay, Argentina, and the Touat Valley in Africa); however these locations are not shown in Figure 1 because, at a global scale, they are not where braided rivers are mainly found." To further justify our selection of locations displayed, we have added the following sentence to the manuscript at L75: "The regions displayed in Figure 1 are regularly cited in literature on braided rivers as the main regions where this river type can be found (e.g., Tockner et al., 2006; Hibbert & Brown, 2001)." We have added Russia to Fig. 1 based on comments in studies (Chalov & Alexeevsky (2015), Alexeevsky et al. (2013)) about the
high number of braided rivers in the country.

In regard to braided rivers in the United States, we respectfully point to L72, where we did
mention the U.S. containing braided rivers. Thank you for pointing to the Wikipedia link
(https://commons.wikimedia.org/wiki/Category:Braided_rivers_in_the_United_States_by_s
tate), however it does not appear that this is a reliable authority on instances of braided rivers
in the U.S. The webpage is a list of user-generated images that have included tags with the
wording "braided river", however many of the images shown here do not appear to be of
braided rivers (e.g. on the pages for Alabama, Virginia, Utah and Massachusetts).

*3. The word "hyporheic" only appears in the abstract and end of the paper, not in the main*
*body. This mention in the abstract should be removed since it is not a topic covered in the*
*paper. It is probably better left to another paper as the issues in measuring hyporheic flow*
*differ significantly.*

Response: Thank you for your comment, however the term "hyporheic" was mentioned 16
times throughout the original manuscript, and it was specifically discussed in conjunction
with several of the cited studies. We believe this review would be significantly lacking if we
did not include information on hyporheic flow processes. As we have noted at L113, issues
surrounding hyporheic zone processes are of great importance in the management of braided
rivers. There is a significant amount of river flow that occurs within the hyporheic zone in
braided rivers. Further, as highlighted at L426, L568, L880 and L946, it is often difficult to
distinguish between regional groundwater discharge into rivers and re-emerging river water
from the hyporheic zone. While this is an issue that would be faced in other river
environments, it is likely that this is more of an issue in braided rivers which 1) have highly
permeable river bed strata, and 2) have a significant amount of river flow that occurs within
the streambed. As Reviewer #1 highlighted (see comment #9), we need to clarify what we
mean by hyporheic flow and the hyporheic zone, as well as better explain the related research
gaps. We have addressed this at L60-65 and L113-114.

*4. The modeling discussion is focused too much on MODFLOW. The description of*
*MODFLOW packages can be found elsewhere and there are other models that incorporate*

*groundwater-surface water interaction that could be discussed. For example, a recent special*

*issue in Groundwater on integrated modeling included a paper on streambed heterogeneity.*

*There is also a recent review paper on modeling gw-sw interaction in Reviews of Geophysics*

*that provides a broader view. The abstract mentions the need for new approaches in*

*modeling, but the paper does not provide sufficient direction to justify this as a conclusion of*

*the paper. The conclusion the models need more data and more sensitivity analysis has been*

*stated many times before.*

Response: In the review we focused on MODFLOW as this is the code that most previous studies have used to model groundwater-surface water exchange in braided rivers. We agree that the description of MODFLOW packages can be found elsewhere and in the revised manuscript we have removed these details. Sentences from L749-760 have been replaced with the following: "Several packages are available in MODFLOW for simulating surface water-groundwater interaction and further details about the application and limitations of these can be found in Brunner et al. (2009, 2010)."

We agree that it would be helpful to include specific recommendations on new approaches to modelling including codes (such as HydroGeoSphere) and methods such as those detailed in

Brunner et al. (2017). We have addressed this in section 4 of the revised manuscript at L955-

L1025.

*5. I was surprised that fiber optic temperature systems (also known as DTS for distributed*

*temperature systems) and geophysics were not discussed. These methods have been*

*mentioned in other reviews and provide broader coverage which might benefit braided*

*streams. I found it odd to bring up thermal imaging for the first time in the discussion section*

*rather than in the review of methods, especially since it is mentioned in the abstract and it is*

*one of the more promising techniques for heterogeneous systems. An example of the benefits*

*of thermal imaging might provide an interesting figure.*

Response: Thank you for your suggestions. We agree that the manuscript would benefit from discussing geophysical techniques and DTS as possible methods to apply in braided rivers. In section 4, we have added more detail on additional methods that can be applied in braided rivers.

*6. On the topic of figures, the figures were lacking in illustrative examples of applications.*
*There was a map, but the other figures were photos or diagrams and didn't show quantitative*
*challenges or opportunities. In other words, I think it would help the readers' understanding*
*to include data figures.*
Response: For the sake of brevity, we have decided not to add additional figures to the
manuscript.
*7. One place that the paper focuses on braided streams is the literature review of methods.*
*The paper summarizes applications in braided streams and the table of methods lists braided*
*stream citations. However, the literature summary sections of the paper are a bit dry. They*
*list highlights of each paper one after another. I think some of these papers could be*
*describing non-braided streams and the reader would not know. This type of literature review*
*needs to be briefer and provide synthesis of issues specific to the problem identified. In*
*addition, a significant number of references (estimated 25% based on the first page of the*
*bibliography) are not readily available literature but reports or theses (typically from NZ).*
*Many readers will not have ready access and the focus on one region is not justified.*
Response: We agree that it would be beneficial to shorten the sections that discuss prior
literature, and we have addressed this in the revised manuscript by deleting sections at L204-
214, 220-229, 256-276, 280-284, 336-352, 360-365, 384-387, 392-407, 412-417, 445-452,
457-459, 467-476, 512-543, 548-564, 585-589, 641-643, 655-656, 661-663, 694-695, 700-
704, 749-760, 768-803, and 808-816.
We endeavoured to make it clear through the introduction that the review focuses on braided
rivers, so that if there were other types of rivers discussed, we would specifically note this.
We have added a sentence to this effect in the introduction at L90: "it is important to note, the
specific rivers discussed in this article are all braided rivers unless otherwise mentioned".
In regard to technical reports and theses that have been cited, the referencing requirements for
the journal's publisher did not stipulate for URLs to be included with these reference types.
However, based on the reviewer's recommendation and for the benefit of readers, we have
included URLs where relevant in the reference list of the revised manuscript. We have also
removed the following references that are not publicly available: Anderson (2004), Davey
(2004), Aitchison-Earl & Ritson (2013), Williams & Aitchison-Earl (2006).

We realise that there is a heavy weighting on studies conducted in New Zealand and this is not intentional by any means. The vast majority of published studies on this topic that we were able to find were based in New Zealand. In fact we chose not to discuss several New

Zealand studies in the methods section as we felt studies from this country were over- represented. Through the process of gathering literature, we did consider possible reasons for the apparent over-representation of New Zealand studies. We considered that search engine results may have been weighted to display New Zealand studies as the authors were based in

New Zealand. To assess whether this was the case, in addition to more general searches, we specifically searched for literature in countries or regions (e.g., Italy) where braided rivers are common. This search method produced some, but not many, additional relevant references.

Further, the majority of search engines used were via scientific indexing sites (e.g., Web of

Science), which to the authors' knowledge, do not tailor search results in the way that Google does. We do acknowledge that we are likely to have missed literature published in languages other than English, but this issue is likely not unique to our review.

*8. It can be difficult to meet the standards of a review article. In the end, I ask myself whether*

*I would give this paper to colleagues to read, or just keep recommending Kalbus et al. or*

*LaBaugh and Rosenberry as review papers on the topic. I do not think there is enough new*

*material here for me to consider this paper to be an update on the earlier papers. If revising,*

*I would recommend a very short review paper, which introduces Table 1 and gives the reader*

*the reference list for readers to select topics on their own (rather than the one line summaries*

*of each paper). The shorter paper also needs to provide the reader with an approach to*

*braided streams that is distinctly different than other streams – this message will take*

*additional synthesis and thus I would consider it to be a new paper rather than a*

*resubmission. Hence, I am recommending rejection and significant redirection for any new*

*submittal.*

Response: We are grateful to the reviewer for their constructive comments. While there have been a number of review papers on surface water-groundwater interaction, none have focused on braided rivers previously and this is the gap we are wanting to address. We would argue that braided rivers have features that are unique enough to warrant a review paper that focuses on this river type specifically. As detailed above, we have revised the introduction to clarify the unique characteristics of braided rivers (L157-173). We have shortened the review of previous studies in Section 3 and increased our use of Table 1 to provide guidance to the reader, as suggested. Also, we have expanded section 4 to enhance the novelty of the paper by suggesting emerging and promising techniques being used in other environments and that are likely to have application in braided rivers.

We have had very positive feedback on this manuscript from numerous groundwater researchers and managers within our networks and we hope that by using the guidance provided by the reviewer we have addressed the reviewer's concerns within the revised manuscript.

lowland stream flows in the Lake Ellesmere area. Christchurch, New Zealand:
Environment Canterbury. Report Number U06/31.

---

## Referee Report (RR1)

**Specific Comments**

58 there are better references to use here (eg Fleckensten et al. 2010, Gonzalez-Pinzon et al. 2015, Magliozzi et al. (2018) etc). Previous review papers have all pointed to the need to address groundwater-surface water exchange with multiple techniques and scales

63 This definition of hyporheic exchange is inconsistent with the traditional definition of previous authors (eg Boano et al. 2014), which refer to hyporheic exchange as being return flow to the river along localised flowpaths. The definition provided here includes unidirectional exchange through the hyporheic zone (also undefined). With this in mind, the text would be greatly improved by dedicating a section to terminology and the difficulties in applying terms to braided rivers. Refer to the review by Magliozzi et al. (2018) which discusses hyporheic exchange at different scales.

107 its worth pointing out that preferential flow paths occur at multiple scales

112-113 this lack of knowledge about hyporheic exchange relates to the lack of conceptual understanding of how braided rivers function in the subsurface, hence definitions are problematic in this setting

114 they may be dry at the surface, but there could still be subsurface (hyporheic) flow between reaches due to sediment heterogeneity and large lateral and downstream elevation changes

133 land encroachment and containment through flood engineering could be added to this list

165 it's the clast-supported gravels which act as preferential flowpaths. OFGs are an extreme type of clast-supported gravel

191-192 I disagree with this statement. The main reason for measuring during low flow conditions is to avoid errors caused by river flow recession. In many rivers the hourly flow recession rate greatly exceeds the rate of leakage to groundwater. By measuring at low flow, the error caused by the time difference between concurrent surveys, and the time difference due to downstream flow propagation are both minimised.

331 Equilibrium radon concentrations in groundwater hosted by alluvial gravels tend to be quite variable because of the spatial variability in mineralogy and water-rock interaction. This variability limits the utility of Rn-222 method

354 The alkalinity section is generic and not specific to braided rivers

390 The temperature section could be improved by treating the data collection via different scales using DTS, heat probes, remote sensing, and time series. With respect to modelling, note that temperature data is extremely difficult to model within an alluvial aquifer because of interference from the surface temperature gradient, and uncertainty around the thermal buffering effect of the gravel medium. Also, because braided rivers and associated gravels are highly dynamic, heat transfer within heterogeneous medium can be highly non-linear.

438 This section should include a discussion on the importance of conceptualisation, since the relationship between rivers and the regional groundwater table can be connected, disconnected, or transitional (see eg Brunner et al 2009)

459-461 The previous comment on conceptualisation relates to how Darcy's law is applied. Are the piezometer measurements in the hyporheic zone, a perched aquifer, or the regional aquifer? Is the river hydraulically connected to either of those aquifers?

504 use the term 'parafluvial hyporheic zone' for consistency

513 The paper could be greatly improved by adding more discussion on measurement scales, particularly when it comes to hydraulic conductivity. Many of the measurement techniques listed in this and prior section pose a challenge of how to upscaling the results for the purposes of quantification. There is a significant challenge to scaling up measurements made in braided rivers.

570-573 Braided river deposits are known to be highly anisotropic because of stratification and particle imbrication. This section could be improved by focusing less on bulk hydraulic property measurements, and focussing more on methods for determining anisotropy since the vertical hydraulic conductivity component is the controlling variable for river leakage.

591-593 I think the only study listed here which is dynamic is Wohling et al 2018 (I could be wrong)

615 The real advantage of modelling is its ability to integrate a diversity of data types at a range of temporal and spatial scales

628 Most of the general comments I've made could be addressed in this section. E.g. the comment referenced to Lovett in 666-667 provides a starting point for discussing the need for improved conceptualisation of braided river hydrology

655-716 This section is applicable to all rivers, apart from 688-691 which is more specific to braided rivers. The paper would be greatly improved by severely trimming this section down and focussing more on issues specific to braided rivers

718 At this stage the paper starts to get messy as it's the third time that some approaches are discussed. This section could be moved to earlier in the paper and revised to focus on the context for measurement approaches. Some approaches can help with conceptual understanding, and some can be used for quantification of exchanges.

720-724 This is the first discussion about conceptual understanding, but a conceptual understanding should be the starting point for knowing which methods to apply, and how to interpret the results. This highlights another opportunity for this paper: many of the methods cannot be used for quantification, but they can help us with process understanding. This distinction is not clear in the paper.

732-773 This section would be better integrated into the relevant measurement sections on measurement methods.

775-787 This section would be better placed in the methods section

789-796 There are a number of modelling problems specific to braided rivers, to list a few:

- River morphology is constantly changing. A transient change in model structure is not a feature captured by groundwater models, although scripting (eg PyFlow) provides the potential for this to be achieved
- The braided river bedform cannot be adequately characterised with the existing Modflow SFR functions
- To simulate the dynamic nature of a subsurface braided river hydrology requires a fully coupled model (alluded to in 789-796)

---

## Author Response (AR2)

[revised manuscript text omitted]

Dear Assoc. Prof. Saco

Thank you very much for organising the reviews of our manuscript. We have responded to all of the referees' comments. With the latest changes to the manuscript, we have addressed Referee #3's comments regarding the need to better understand the subsurface of braided rivers and methods that can be used to do so. We have added comments in the manuscript in this regard at L43, L67-70, L155, L455-457, L658-659, L739, L760, L800-801, and L854-856.

**Referee #3:** Please see separate uploaded document.

**Referee #4:**

*The manuscript gives a nice review on methods measuring groundwater–surface water exchange in braided rivers. Perhaps I would argue that the review is wider than just for braided rivers, but still very useful. I think that the authors have done very good job in the revisions. I believe that the revised manuscript is greatly improved by considering the reviewer comments.*

Response: Thank you very much for your review and positive feedback. We are pleased to hear that our revisions have been helpful in improving the manuscript.

**Reply to Reviewer # 3**

We thank Reviewer #3 for the helpful review and have responded to each comment below.

All line numbers in our responses refer to the revised marked manuscript.

**General Comments**

*1.        The paper reviews methods for the quantification of braided river exchanges, and*

*discusses various quantification approaches and their pros and cons. The paper could be*

*improved by emphasizing methods that can be used for characterization and process*

*understanding, as well as those used for quantification. The main difficulty with braided river*

*studies may not be the difficulty in taking measurements as much as the difficulty in*

*interpreting what the measurement results actually mean. The approaches to quantification*

*that have been outlined by the authors do require a physical context in order to interpret the*

*values measured after all.*

Response: Many thanks for the insightful and constructive comments. We have endeavoured to add various text to the revised manuscript in line with your suggestions here. These are detailed in the comment responses below.

*2.        It follows that a key subject not covered by the authors is process and conceptual*

*understanding of subsurface hydrology of braided rivers. This has implications for the*

*application of different methods and interpretation of the data derived, as well as the use of*

*hydrological terminology within braided river settings. For example, how does a practitioner*

*define the hyporheic zone or hyporheic exchange for a braided river? The problem stems*

*from there being very few studies on braided river subsurface hydrology and*

*conceptualisation (Huber and Huggenberger 2016 is a notable exception). Addressing the*

*conceptualisation of subsurface braided river hydrology, or lack of in the literature, provides*

*the authors an opportunity to highlight the potential for characterization methods (eg*

*geophysics, remote sensing) to improve our understanding of how braided rivers could work*

*in the subsurface. This in turn helps us to interpret field measurements and clarifies our*

*terminology.*

Response: As discussed further in the comments below, we have clarified our definitions around the hyporheic zone and flow at L62-70 in the revised manuscript. We have added comments in the manuscript relating to the need to improve our conceptual understanding of braided rivers (particularly their subsurface) and methods that may help do this (e.g., geophysics) at L43, L67-70, L155, L455-457, L658-659, L739, L760, L800-801, and L854-856.

*3.    A paper focused more specifically on braided rivers could be achieved via a restructuring to bring issues/knowledge gaps in braided rivers to the forefront, condensing of the descriptions for individual methods, and moving much of the current discussion to the relevant methods sections.*

Response: Thank you for your comment. We have condensed the discussion section, as discussed further in Comment #23 below, to be tailored more specifically to braided rivers. As also noted below, we have decided not to make significant structural changes to the manuscript as you have suggested in light of the Editor's recommendation for minor changes only.

**Specific Comments**

*4.    L58 there are better references to use here (eg Fleckensten et al. 2010, Gonzalez-Pinzon et al. 2015, Magliozzi et al. (2018) etc). Previous review papers have all pointed to the need to address groundwater-surface water exchange with multiple techniques and scales.*

Response: Thank you for the excellent suggestion for more appropriate references here. We have added the references you have suggested (Fleckenstein et al., 2010; González-Pinzón et al., 2015; Magliozzi et al., 2018) and removed the reference to Lovett et al. (2015).

*5.    L63 This definition of hyporheic exchange is inconsistent with the traditional definition of previous authors (eg Boano et al. 2014), which refer to hyporheic exchange as being return flow to the river along localised flowpaths. The definition provided here includes unidirectional exchange through the hyporheic zone (also undefined). With this in mind, the text would be greatly improved by dedicating a section to terminology and the difficulties in applying terms to braided rivers. Refer to the review by Magliozzi et al. (2018) which discusses hyporheic exchange at different scales.*

Response: Thank you for your comments here, as we recognise that definitions regarding hyporheic exchange are important to clarify. We have amended this paragraph to be more in line with the Boano et al. (2014) definition of hyporheic flow and the hyporheic zone. At
L66, we have noted that hyporheic flow is multidirectional and occurs at various spatial and
temporal scales. With respect, and with brevity in mind, we feel that these amendments
clarify the terminology around hyporheic flows sufficiently for the purposes of this paper.

6.    *L107 its worth pointing out that preferential flow paths occur at multiple scales.*
Response: We have added a comment to this effect in this sentence.

7.    *L112-113 this lack of knowledge about hyporheic exchange relates to the lack of*
*conceptual understanding of how braided rivers function in the subsurface, hence definitions*
*are problematic in this setting.*
Response: Note in the response to Comment #2 above, we have added text relating to the lack
of conceptual understanding of the subsurface of braided rivers elsewhere in the manuscript
and feel the text here now has sufficient supporting context.

8.    *L114 they may be dry at the surface, but there could still be subsurface (hyporheic)*
*flow between reaches due to sediment heterogeneity and large lateral and downstream*
*elevation changes.*
Response: Indeed, this may be the case. We have amended the sentence to read: "Braided
rivers also often have reaches that become dry or have very low flows at the surface."

9.    *L133 land encroachment and containment through flood engineering could be added*
*to this list.*
Response: We had considered these included in this list, but we agree, that it would be
helpful to explicitly state these factors. The sentence has been amended in the revised
manuscript at L137-138.

10.   *L165 it's the clast-supported gravels which act as preferential flowpaths. OFGs are*
*an extreme type of clast-supported gravel.*
Response: While this may indeed be the case, the literature we have come across specifically
looking at preferential flowpaths in alluvial aquifers deposited by gravel braided rivers refers
to this OFG terminology, so we would prefer to keep the wording as is to stay in line with
existing braided river literature.

*11.     L191-192 I disagree with this statement. The main reason for measuring during low*
*flow conditions is to avoid errors caused by river flow recession. In many rivers the hourly*
*flow recession rate greatly exceeds the rate of leakage to groundwater. By measuring at low*
*flow, the error caused by the time difference between concurrent surveys, and the time*
*difference due to downstream flow propagation are both minimised.*
Response: Your comment has highlighted the need to clarify the wording here, as our
intention was the same as yours, i.e., to indicate that river flow recession will often cause
errors in measuring gw-sw exchange. We have amended the wording as follows at L197-198:
"Measurements should generally be taken in low flow conditions to avoid errors caused by
river flow recession after rainfall or snowmelt."
*12.     L331 Equilibrium radon concentrations in groundwater hosted by alluvial gravels*
*tend to be quite variable because of the spatial variability in mineralogy and water-rock*
*interaction. This variability limits the utility of Rn-222 method.*
Response: Thank you for your comment. We have added a note around Rn-222
concentrations being affected by variations in mineralogy of the rocks at L340-341. As for
effects of water-rock interaction, we feel this is covered by the assumptions listed in L335-
338.
*13.     L354 The alkalinity section is generic and not specific to braided rivers.*
Response: This section was included because this method was used in braided river settings
(in the River Feshie catchment). We have added a note at L362 to clarify that this is a braided
river setting.
*14.     L390 The temperature section could be improved by treating the data collection via*
*different scales using DTS, heat probes, remote sensing, and time series. With respect to*
*modelling, note that temperature data is extremely difficult to model within an alluvial*
*aquifer because of interference from the surface temperature gradient, and uncertainty*
*around the thermal buffering effect of the gravel medium. Also, because braided rivers and*
*associated gravels are highly dynamic, heat transfer within heterogeneous medium can be*
*highly non-linear.*
Response: Thank you for the suggestion, however at this stage we prefer not to restructure
this section. We discuss the different scales of methods in L404-413 and L434-437. In regard
to the effects of the surface temperature gradient and uncertainties around thermal buffering of gravel, we have chosen not to add comments to this effect in the revised manuscript because we are not aware of relevant references to support these comments. We appreciate your final comment here and have added at L443-444: "In addition, due to the dynamic nature of braided rivers and their associated sediments, heat transfer within the heterogeneous materials may be non-linear."

15.     *L438 This section should include a discussion on the importance of conceptualisation, since the relationship between rivers and the regional groundwater table can be connected, disconnected, or transitional (see eg Brunner et al 2009).*
Response: Thank you for highlighting this. We have added the following sentence at L455-457: "It is worthwhile to note that it is important to obtain a conceptual understanding of the relationship of the river to the groundwater table, as the river might be connected, disconnected or in a transitional state between the two (Brunner et al., 2009)."

16.     *L459-461 The previous comment on conceptualisation relates to how Darcy's law is applied. Are the piezometer measurements in the hyporheic zone, a perched aquifer, or the regional aquifer? Is the river hydraulically connected to either of those aquifers?*
Response: Where the piezometers are installed and the relationship between the river and adjacent aquifer would be study specific. We feel the comment added to L455-457 sufficiently highlights the need to consider this conceptualisation.

17.     *L504 use the term 'parafluvial hyporheic zone' for consistency.*
Response: Thank you, we have amended this sentence accordingly.

18.     *L513 The paper could be greatly improved by adding more discussion on measurement scales, particularly when it comes to hydraulic conductivity. Many of the measurement techniques listed in this and prior section pose a challenge of how to upscaling the results for the purposes of quantification. There is a significant challenge to scaling up measurements made in braided rivers.*
Response: We have discussed issues of scaling up measurements in L567-587 in relation to hydraulic conductivity. We have also discussed issues of scale and possible ways to overcome them throughout the paper in relation the various methods discussed.

19. *L570-573 Braided river deposits are known to be highly anisotropic because of stratification and particle imbrication. This section could be improved by focusing less on bulk hydraulic property measurements, and focussing more on methods for determining anisotropy since the vertical hydraulic conductivity component is the controlling variable for river leakage.*

Response: Thank you for the suggestion. We have added to the end of this section, at L584-587, the following: "Arguably, as vertical hydraulic conductivity is the controlling factor for river losses, there should be more focus on estimating anisotropy values of the braided river substrate. Methods for estimating anisotropy have been demonstrated using aquifer tests (Neuman et al., 1984; Mutch, 2005; Mathias and Butler, 2007) and more recently geophysics (Al-Hazaimay et al., 2016; Fernández-Álvarez et al., 2016)."

20. *L591-593 I think the only study listed here which is dynamic is Wohling et al 2018 (I could be wrong).*

Response: The wording in the revised manuscript has been changed at L606 from "flow dynamics and pumping impacts" to "groundwater-surface water interactions" to avoid any confusion.

21. *L615 The real advantage of modelling is its ability to integrate a diversity of data types at a range of temporal and spatial scales.*

Response: Thank you for highlighting this, and we agree. We have added at L633-634: "It is also possible to integrate a range of data types at varying spatial and temporal scales with modelling."

22. *L628 Most of the general comments I've made could be addressed in this section. E.g. the comment referenced to Lovett in 666-667 provides a starting point for discussing the need for improved conceptualisation of braided river hydrology.*

Response: We have attempted to address your general comments throughout the manuscript so that the first mention of the need for improved conceptualisation is at the beginning of the paper and mentioned throughout. In this section specifically, we have at L658-659 added a note on the value of qualitative data to improve conceptual understanding.

*23.     L655-716 This section is applicable to all rivers, apart from 688-691 which is more specific to braided rivers. The paper would be greatly improved by severely trimming this section down and focussing more on issues specific to braided rivers.*

Response: This section has now been significantly cut down and tailored as specifically as possible to braided rivers.

*24.     L718 At this stage the paper starts to get messy as it's the third time that some approaches are discussed. This section could be moved to earlier in the paper and revised to focus on the context for measurement approaches. Some approaches can help with conceptual understanding, and some can be used for quantification of exchanges.*

Response: Thank you for your comments. With the deletion of text mentioned in Comment #23 above, the repetition of methods discussion has been significantly reduced and we feel this section does not need to be moved to avoid repetition. We have added a note at L800-801 regarding geophysics being a promising method for characterising the subsurface of braided rivers. There is also a mention at L769 of the use of tracers for investigating hyporheic zone flow.

*25.     L720-724 This is the first discussion about conceptual understanding, but a conceptual understanding should be the starting point for knowing which methods to apply, and how to interpret the results. This highlights another opportunity for this paper: many of the methods cannot be used for quantification, but they can help us with process understanding. This distinction is not clear in the paper.*

Response: Refer to Comment #2 above in regard to mentions of conceptual understanding of braided rivers that have been added throughout the manuscript. We have added a sentence at L854-856 noting that while only some methods can be used for quantification, many methods can be used for improving conceptual understandings.

*26.     L732-773 This section would be better integrated into the relevant measurement sections on measurement methods.*

Response: Thank you for the suggestion, however given the Editor's request to make minor changes only, we would prefer not to restructure the manuscript at this stage.

*27.     L775-787 This section would be better placed in the methods section.*

Response: Thank you for the suggestion, however given the Editor's request to make minor
changes only, we would prefer not to restructure the manuscript at this stage.

*28.* *L789-796 There are a number of modelling problems specific to braided rivers, to list*
*a few:*

• *River morphology is constantly changing. A transient change in model structure is not*
*a feature captured by groundwater models, although scripting (eg PyFlow) provides*
*the potential for this to be achieved.*

• *The braided river bedform cannot be adequately characterised with the existing*
*Modflow SFR functions.*

• *To simulate the dynamic nature of a subsurface braided river hydrology requires a*
*fully coupled model (alluded to in 789-796).*

Response: Thank you for these suggestions, we have added these comments to this section in
the revised manuscript at L813-814 and L818-821.